# On the Parameter Identifiability of Partially Observed Linear Causal Models

**Xinshuai Dong**[1]*    **Ignavier Ng**[1]*    **Biwei Huang**[2]    **Yuewen Sun**[3]    **Songyao Jin**[3]
**Roberto Legaspi**[4]    **Peter Spirtes**[1]    **Kun Zhang**[1,3]
[1]Carnegie Mellon University  [2]University of California San Diego
[3]Mohamed bin Zayed University of Artificial Intelligence  [4]KDDI Research

## Abstract

Linear causal models are important tools for modeling causal dependencies and yet in practice, only a subset of the variables can be observed. In this paper, we examine the parameter identifiability of these models by investigating whether the edge coefficients can be recovered given the causal structure and partially observed data. Our setting is more general than that of prior research—we allow all variables, including both observed and latent ones, to be flexibly related, and we consider the coefficients of all edges, whereas most existing works focus only on the edges between observed variables. Theoretically, we identify three types of indeterminacy for the parameters in partially observed linear causal models. We then provide graphical conditions that are sufficient for all parameters to be identifiable and show that some of them are provably necessary. Methodologically, we propose a novel likelihood-based parameter estimation method that addresses the variance indeterminacy in a specific way and can asymptotically recover the underlying parameters up to trivial indeterminacy. Empirical studies on both synthetic and real-world datasets validate our identifiability theory and the effectiveness of the proposed method in the finite-sample regime. Code: `https://github.com/dongxinshuai/scm-identify`.

## 1 Introduction and Related Work

Causal models, which serve as a fundamental tool to capture causal relations among random variables, have achieved great success in many fields [49, 39, 40, 44]. A fundamental problem in the field is how and to what extent can we identify the underlying causal model given observational data. When all variables are observed, the problem has been well studied: the underlying structure can be identified up to the Markov equivalence class, e.g., by the PC [49] or GES [13] algorithm; when the structure is given, the causal coefficient (direct causal effect) between two variables can also be identified [8, 39].

However, in real-world systems, the variables of interest may only be partially observed. Thus, considerable efforts have been dedicated to identification of causal models in the presence of latent variables. One line of research focuses on structure learning given partially observed variables. Notable approaches include FCI and its variants [49, 38, 14, 2], as well as ICA-based [23, 43], tetrad-based [48, 28], high-order moments-based [46, 11, 58, 1, 12], and rank constraint-based [48, 24, 18] methods.

In this paper, we focus on the the identification of parameters of a partially observed model. Specifically, given the causal structure of and observational data from a partially observed causal model, we are interested in identifying all the parameters, and thus the underlying causal model can be fully specified. To identify the parameters, a classical way is to project the directed acyclic graph (DAG) with latent variables to an acyclic directed mixed graph (ADMG) or partially ancestral graph [42], without explicitly modeling the latent confounders. Based on ADMG, graphical criteria such as half-trek [20], G-criterion [9], and some further developments [51, 29] have been proposed to establish the parameter identifiability. Another way is to leverage do-calculus, proxy variables, and instrumental

---

*Equal contribution.
38th Conference on Neural Information Processing Systems (NeurIPS 2024).

variables [47, 39, 25] to identify the direct causal effect, which corresponds to the edge coefficient in linear causal models. For a more detailed discussion of related work, please refer to Appendix D.

Despite the effectiveness of current methods for parameter identification, however, they have two main drawbacks: they require all the variables to be connected in specific ways, and only focus on identifying the edge coefficients between observed variables. To this end, in this paper we propose a novel framework that considers a more general setting for parameter identification. To be specific, we allow all variables, including both observed and latent ones, to be flexibly related, and we aim to recover the edge coefficients among all variables, even including those from a latent variable to another latent variable or another observed variable, which previous methods cannot handle. We summarize our contributions as follows.

- To the best of our knowledge, we are the first to consider parameter identifiability of partially observed causal model in the most general scenario—all variables, including both observed and latent ones, are allowed be flexibly related, and edge coefficients between any pair of variables are concerned. In contrast, most existing works consider only the edges between observed variables.
- Theoretically, we identify three types of parameter indeterminacy in partially observed linear causal models. We then provide graphical conditions that are sufficient for all parameters to be identifiable and show that some of them are provably necessary. These necessary conditions also offer insights into scenarios where the parameters are guaranteed to be non-identifiable.
- Methodologically, we propose a novel likelihood-based parameter estimation method, which parameterizes population covariance in specific ways to address variance indeterminacy. Our empirical studies on both synthetic and real-world data validate the effectiveness of our method in the finite-sample regime, even under certain misspecification of the underlying causal model.

## 2 Preliminaries

### 2.1 Problem Setting

In this work, we focus on partially observed linear causal models, defined as follows.

**Definition 1** (Partially Observed Linear Causal Models). *Let $\mathcal{G} := (\mathbf{V}_\mathcal{G}, \mathbf{E}_\mathcal{G})$ be a DAG. Each variable $V_i \in \mathbf{V}_\mathcal{G}$ follows a linear structural equation model $\mathsf{V}_i = \sum_{\mathsf{V}_j \in Pa_\mathcal{G}(\mathsf{V}_i)} f_{j,i} \mathsf{V}_j + \epsilon_{\mathsf{V}_i}$, where $\mathbf{V}_\mathcal{G} := \mathbf{L}_\mathcal{G} \cup \mathbf{X}_\mathcal{G} = \{\mathsf{L}_i\}_{i=1}^m \cup \{\mathsf{X}_i\}_{i=m+1}^{m+n}$ contains $m$ latent variables and $n$ observed variables. $Pa_\mathcal{G}(\mathsf{V}_i)$ denotes the parent set of $\mathsf{V}_i$, $f_{j,i}$ denotes the edge coefficient from $V_j$ to $V_i$, and $\epsilon_{\mathsf{V}_i}$ represents the Gaussian noise term of $\mathsf{V}_i$.*

We drop the subscript $\mathcal{G}$ in $\mathbf{L}_\mathcal{G}$ and $\mathbf{X}_\mathcal{G}$ when the context is clear. We use $\mathsf{V}$, $\mathbf{V}$, and $\mathcal{V}$ to denote a random variable, a set of variables, and a set of sets of variables, respectively. In Definition 1, the relations between variables can also be written in the matrix form as $\mathbf{V}_\mathcal{G} = F^T \mathbf{V}_\mathcal{G} + \epsilon_{\mathbf{V}_\mathcal{G}}$, where $F = (f_{j,i})_{i,j \in [m+n]}$ is the weighted adjacency matrix. Here, $f_{j,i} \neq 0$ if and only if $V_j$ is a parent of $V_i$ in $\mathcal{G}$. We also write

$$F = \begin{matrix} & \mathbf{L} & \mathbf{X} \\ \begin{matrix} \mathbf{L} \\ \mathbf{X} \end{matrix} & \begin{pmatrix} F_{\mathbf{LL}} & F_{\mathbf{LX}} \\ F_{\mathbf{XL}} & F_{\mathbf{XX}} \end{pmatrix} \end{matrix} \quad \text{and} \quad \Omega = \begin{pmatrix} \Omega_{\epsilon_\mathbf{L}} & 0 \\ 0 & \Omega_{\epsilon_\mathbf{X}} \end{pmatrix},$$

where $\Omega$ is the diagonal covariance matrix of $\epsilon_{\mathbf{V}_\mathcal{G}}$.

Our objective is to identify $F$, the causal edge coefficients of the model, given observational data and the causal structure $\mathcal{G}$. Denote by $\Sigma_\mathbf{L}$ and $\Sigma_\mathbf{X}$ the population covariance matrix of latent variables $\mathbf{L}$ and observed variables $\mathbf{X}$, respectively; their precise formulations are provided in Proposition 1. We also denote by $\sigma_{i,j}$ the $(i,j)$-th entry of $\Sigma_\mathbf{X}$. In this work, we assume that the noise terms of latent variables, $\epsilon_\mathbf{L}$, have unit variance, i.e., $\Omega_{\epsilon_\mathbf{L}} = I$, which will be justified later in Section 3.1. Note that variables are partially observed and thus we only have access to i.i.d. samples of observed variables $\mathbf{X}$. As variables are jointly Gaussian, the observations can asymptotically be summarized as the population covariance matrix $\Sigma_\mathbf{X}$. In other words, we aim to identify $F$ and $\Omega$ given $\Sigma_\mathbf{X}$ and $\mathcal{G}$. The identification of parameters is important in that, once we identify the parameters, the underlying causal model is fully specified, and thus we can flexibly calculate causal effects, infer interventional distributions, and finally answer counterfactual queries [39]. It is worth noting that, for parameter identification, the structure $\mathcal{G}$ is assumed to be known, which is different from the setting of causal discovery where the goal is to identify $\mathcal{G}$ from data.

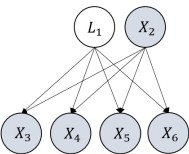 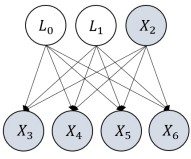 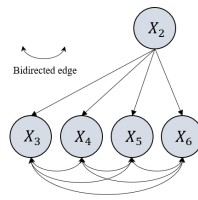

(a) Graph $\mathcal{G}_1$. Its parameters can be fully identifiable.

(b) Graph $\mathcal{G}_2$. Its parameters cannot be fully identified.

(c) $\mathcal{G}_1$ $\mathcal{G}_2$ have the same ADMG in the latent projection framework.

Figure 1: Illustrations of the advantage of our framework. Within our framework, it can be shown that $\mathcal{G}_1$'s parameters can be identified (up to sign) while $\mathcal{G}_2$'s cannot. In contrast, the latent projection framework cannot even differentiate $\mathcal{G}_1$ from $\mathcal{G}_2$ as they share the same ADMG (c) after projection. Furthermore, with ADMG, any edge coefficient that involves a latent variable cannot be considered.

## 2.2 Framework Comparison

Without latent variables, it has been shown all parameters are identifiable [8]. However, the problem becomes very challenging when latent variables exist. There are two lines of research. One focuses on the use of do-calculus, proxy variables, and instrumental variables to identify direct causal effects among observed variables [47, 39, 25] (in linear models the direct causal effect is captured by the edge coefficient). Another line addresses latent confounders by projecting a DAG with latent variables into an ADMG, where the confounding effects of latent variables are simplified and represented by correlation among noise terms [20, 9, 51, 29]. An example is in Figure 1, where (a) is the original graph and (c) is the projected ADMG whose bidirected edges correspond to correlated noise terms.

Compared to the two previous lines of thought, our framework has two advantages. To begin with, we additionally considers the identifiability of coefficients of edges that involve latent variables. For example, in Figure 1, we aim to identify all the coefficients including the one from $L_1$ to $X_3$, i.e., $f_{1,3}$. In contrast, the proxy variable framework and the latent projection framework identify only the coefficients among observed variables: the proxy variable framework focuses only on the direct causal effect from one observed variable to another observed variable, while the latent projection framework transforms all latent variables into bidirected edges and thus can never identify the coefficient of the edge that has a latent variable as the head or tail.

Furthermore, the projection framework deals with latent variables in a rather brute-force way: dense latent confounding effects among observed variables may be caused by only a small number of latent variables, but that information is lost during projection. For example, in Figure 1, (a) and (b) share the same ADMG after projection, i.e., (c). However, as we will show later, parameters in (a) can be identified, while in (b) the parameters cannot. If we only consider the ADMG in (c), then we can never capture this nuance and thus cannot identify the coefficients that we might be able to.

# 3 Identifiability Theory

## 3.1 Definition of Parameter Identifiability and Indeterminacy

We follow the notion of generic identifiability and define parameter identifiability as follows.

**Definition 2** (Identifiability of Parameters of Partially Observed Linear Causal Models). *Let $\theta = (F, \Omega) \in \Theta$. We say that $\theta$ is generically identifiable, if the mapping $\phi(\theta) = \Sigma_{\mathbf{X}}$ is injective, for almost all $\theta \in \Theta$ with respect to the Lebesgue measure.*

Definition 2 indicates if parameter $\theta$ is identifiable, then there does not exist $\theta' \in \Theta$ that entails the same observations as those of $\theta$. As in the typical literature of parameter identification, we consider generic identifiability to rule out some rare cases where the parameters for that structure is generally identifiable, but with some specific parameterization, the parameters cannot be identified. This is similar to faithfulness in causal discovery [49] and we will provide an example in Example 1. We next introduce three important indeterminacies about parameter identification when latent variables exist.

**Theorem 1** (Indeterminacy of Scaling of $\Omega_{\epsilon_{\mathbf{L}}}$). *Consider a model that follows Def. 1 with number of latent variables $m \geq 1$ and $\theta = (F_{\mathbf{LL}}, F_{\mathbf{LX}}, F_{\mathbf{XL}}, F_{\mathbf{XX}}, \Omega_{\epsilon_{\mathbf{L}}}, \Omega_{\epsilon_{\mathbf{X}}})$. Let $\Lambda$ be any invertible diagonal matrix, and $\tilde{\theta} = (\tilde{F}_{\mathbf{LL}}, \tilde{F}_{\mathbf{LX}}, \tilde{F}_{\mathbf{XL}}, \tilde{F}_{\mathbf{XX}}, \tilde{\Omega}_{\epsilon_{\mathbf{L}}}, \tilde{\Omega}_{\epsilon_{\mathbf{X}}})$, where*

$$\tilde{F}_{\mathbf{LL}} = \Lambda^{-1} F_{\mathbf{LL}} \Lambda, \ \tilde{F}_{\mathbf{LX}} = \Lambda^{-1} F_{\mathbf{LX}}, \ \tilde{F}_{\mathbf{XL}} = F_{\mathbf{XL}} \Lambda, \ \tilde{F}_{\mathbf{XX}} = F_{\mathbf{XX}}, \ \tilde{\Omega}_{\epsilon_{\mathbf{L}}} = \Lambda^2 \Omega_{\epsilon_{\mathbf{L}}}, \ \tilde{\Omega}_{\epsilon_{\mathbf{X}}} = \Omega_{\epsilon_{\mathbf{X}}}.$$

*Then, $\tilde{\theta}$ and $\theta$ entail the same observations, i.e., $\tilde{\Sigma}_{\mathbf{X}} = \Sigma_{\mathbf{X}}$. Furthermore, we have $\tilde{\Sigma}_{\mathbf{L}} = \Lambda \Sigma_{\mathbf{L}} \Lambda$.*

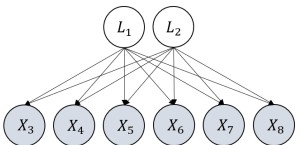 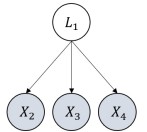 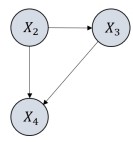

(a) $\mathcal{G}_1$. Its structure is identifiable but its parameters are not identifiable even if the structure is given (due to orthogonal indeterminacy).

(b) $\mathcal{G}_2$. Its structure is not identifiable but params are identifiable.

(c) $\mathcal{G}_3$. $\mathcal{G}_2$'s structure is not identifiable due to the existence of $\mathcal{G}_3$.

Figure 2: Illustrative examples to show that the graphical condition for structure-identifiability and parameter-identifiability could be very different.

A similar theoretical result is provided in [4], and yet our setting is much more general and takes that of [4] as a special case: in our setting, all variables including latent and observed ones can be arbitrarily related while in [4] latent variables cannot be the effect of observed variables.

**Remark 1** (Implication of Theorem 1). *A key implication of Theorem 1 is that, without further assumption, the edge coefficients involving latent variables, i.e., $(F_{\mathbf{LL}}, F_{\mathbf{LX}}, F_{\mathbf{XL}})$, can never be identified, as there always exists a diagonal matrix $\Lambda$ such that $\tilde{\theta}$ and $\theta$ entail the same observations but $(\tilde{F}_{\mathbf{LL}}, \tilde{F}_{\mathbf{LX}}, \tilde{F}_{\mathbf{XL}}) \neq (F_{\mathbf{LL}}, F_{\mathbf{LX}}, F_{\mathbf{XL}})$. Thus, in the rest of this paper, we assume that the noise terms of latent variables, $\epsilon_{\mathbf{L}}$, have unit variance, i.e., $\Omega_{\epsilon_{\mathbf{L}}} = I$. Under this assumption, we have $(\tilde{\Omega}_{\epsilon_{\mathbf{L}}})_{i,i} = \Lambda_{i,i}^2 (\Omega_{\epsilon_{\mathbf{L}}})_{i,i} = 1, i \in [m]$, which implies $\Lambda_{i,i} = \pm 1$. As such, this assumption makes parameter identifiability possible. However, even though we fix the scaling of $\Omega_{\epsilon_{\mathbf{L}}}$, there still exists indeterminacy about the sign of parameters, captured by Theorem 2.*

**Theorem 2** (Group Sign Indeterminacy). *Consider a model that follows Def. 1 with number of latent variables $m \geq 1$, $\theta = (F_{\mathbf{LL}}, F_{\mathbf{LX}}, F_{\mathbf{XL}}, F_{\mathbf{XX}}, \Omega_{\epsilon_{\mathbf{L}}}, \Omega_{\epsilon_{\mathbf{X}}})$, and $\Omega_{\epsilon_{\mathbf{L}}} = I$. Let $S$ be a diagonal sign matrix (entries are either $1$ or $-1$), and $\tilde{\theta} = (\tilde{F}_{\mathbf{LL}}, \tilde{F}_{\mathbf{LX}}, \tilde{F}_{\mathbf{XL}}, \tilde{F}_{\mathbf{XX}}, \tilde{\Omega}_{\epsilon_{\mathbf{L}}}, \tilde{\Omega}_{\epsilon_{\mathbf{X}}})$, where*

$$\tilde{F}_{\mathbf{LL}} = SF_{\mathbf{LL}}S, \ \tilde{F}_{\mathbf{LX}} = SF_{\mathbf{LX}}, \ \tilde{F}_{\mathbf{XL}} = F_{\mathbf{XL}}S, \ \tilde{F}_{\mathbf{XX}} = F_{\mathbf{XX}}, \ \tilde{\Omega}_{\epsilon_{\mathbf{L}}} = \Omega_{\epsilon_{\mathbf{L}}} = I, \ \tilde{\Omega}_{\epsilon_{\mathbf{X}}} = \Omega_{\epsilon_{\mathbf{X}}}.$$

*Then, $\tilde{\theta}$ and $\theta$ entail the same observations, i.e., $\tilde{\Sigma}_{\mathbf{X}} = \Sigma_{\mathbf{X}}$, and $(\tilde{\Sigma}_{\mathbf{L}})_{ii} = (\Sigma_{\mathbf{L}})_{ii}, \ \forall i \in [m]$.*

**Remark 2** (Remark on Theorem 2). *The indeterminacy described in Theorem 2 is referred to as group sign indeterminacy for the following reason: According to the theorem, flipping the sign of $S_{i,i}$ is equivalent to flipping the signs of all coefficients of edges involving the latent variable $\mathsf{L}_i$. This transformation preserves the resulting observations $\Sigma_{\mathbf{X}}$. In essence, each group consists of coefficients of edges involving a particular latent variable.*

**Example 1** (Example for Group Sign Indeterminacy and Generic Identifiability). *In Figure 2 (b), given the structure and $\Sigma_{\mathbf{X}}$, by assuming $\Omega_{\epsilon_{\mathbf{L}}} = I$, the parameters are generally identifiable up to group sign indeterminacy. Specifically, there exist three equality constraints with three free parameters: $f_{1,2}f_{1,3} = \sigma_{2,3}$, $f_{1,2}f_{1,4} = \sigma_{2,4}$, and $f_{1,3}f_{1,4} = \sigma_{3,4}$. The solutions are: (i) $f_{1,2} = \sqrt{\frac{\sigma_{2,3}\sigma_{2,4}}{\sigma_{3,4}}}$, $f_{1,3} = \sigma_{2,3}/f_{1,2}$, $f_{1,4} = \sigma_{2,4}/f_{1,2}$ and (ii) $f_{1,2} = -\sqrt{\frac{\sigma_{2,3}\sigma_{2,4}}{\sigma_{3,4}}}$, $f_{1,3} = -\sigma_{2,3}/f_{1,2}$, $f_{1,4} = -\sigma_{2,4}/f_{1,2}$. The two solutions are different only in terms of group sign. However, if we set $f_{1,2} = 0$, then the parameters are not identifiable (as we will encounter division where the divisor is zero). These rare cases of parameters are of zero Lebesgue measure so we rule out these cases for the definition of identifiability, as in Definition 2.*

Intuitively speaking, group sign indeterminacy arises because one may multiply the latent variable $\mathsf{L}_i$ by $-1$ and accordingly flip the signs of all edge coefficients involving $\mathsf{L}_i$. Note that such an indeterminacy is rather minor for the following reason. (i) In practice, we can always anchor the sign of some edges according to our preference or prior knowledge in order to eliminate the group sign indeterminacy. For example, in Figure 4, if we expect that L2 should be understood as Extraversion instead of non-Exterversion, we can add one additional constraint during our parameter estimation such that the edge coefficient from L2 to E1 ("I am the life of party.") will be positive (as we believe E1 should be positively related to Extraversion). (ii) On the other hand, there are some application scenarios that are not influenced by the group sign indeterminacy, such as causal effect estimations between certain variables. We note that, as the indeterminacy of group sign is rather minor, in the following if the parameters are identifiable only up to group sign indeterminacy, we still say that the parameters are identifiable.

**Definition 3** (Orthogonal Transformation Indeterminacy). *Consider a model that follows Def. 1 with number of latent variables $m \geq 1$, $\theta = (F_{\mathbf{LL}}, F_{\mathbf{LX}}, F_{\mathbf{XL}}, F_{\mathbf{XX}}, \Omega_{\epsilon_\mathbf{L}}, \Omega_{\epsilon_\mathbf{X}})$, and $\Omega_{\epsilon_\mathbf{L}} = I$. We say that there exists an orthogonal transformation indeterminacy in the identification of parameters if there exists a non-diagonal orthogonal matrix $Q$ such that $(F_{\mathbf{LL}}, F_{\mathbf{LX}}, F_{\mathbf{XL}}, F_{\mathbf{XX}}, \Omega_{\epsilon_\mathbf{L}}, \Omega_{\epsilon_\mathbf{X}})$ and $(\tilde{F}_{\mathbf{LL}}, \tilde{F}_{\mathbf{LX}}, \tilde{F}_{\mathbf{XL}}, \tilde{F}_{\mathbf{XX}}, \tilde{\Omega}_{\epsilon_\mathbf{L}}, \tilde{\Omega}_{\epsilon_\mathbf{X}})$ share the same support and entail the same observations, where*

$$\tilde{F}_{\mathbf{LL}} = Q^T F_{\mathbf{LL}} Q, \ \tilde{F}_{\mathbf{LX}} = Q^T F_{\mathbf{LX}}, \ \tilde{F}_{\mathbf{XL}} = F_{\mathbf{XL}} Q, \ \tilde{F}_{\mathbf{XX}} = F_{\mathbf{XX}}, \ \tilde{\Omega}_{\epsilon_\mathbf{L}} = \Omega_{\epsilon_\mathbf{L}} = I, \ \tilde{\Omega}_{\epsilon_\mathbf{X}} = \Omega_{\epsilon_\mathbf{X}}.$$

The orthogonal transformation indeterminacy is the major indeterminacy we consider in the presence of latent variables. Such an indeterminacy also arises in factor analysis [45, 7], which can be viewed as a special case of the data generating procedure considered in Definition 1. Here we only give the definition and will later provide Theorem 4 with an example that captures the scenarios where such indeterminacy exists.

It is worth noting that the graphical condition for structure identifiability and parameter identifiability could be very different. For example, $\mathcal{G}_1$ in Figure 2 (a) is structure-identifiable, and yet the parameters are not identifiable even if the structure is given. In contrast $\mathcal{G}_2$ in Figure 2 (b) is not structure-identifiable, as there exists another structure $\mathcal{G}_3$ in Figure 2 (c) such that $\mathcal{G}_2$ and $\mathcal{G}_3$ can never be differentiated from observational distribution; and yet if $\mathcal{G}_2$ is given, its parameters are identifiable (as in Example 1). Therefore, in this paper, we first consider the cases where the structure can be identified and then study which further conditions are needed for the identifiability of parameters. This will give rise to conditions under which the whole causal model can be fully specified.

## 3.2 Graphical Condition for Structure Identifiability

To explore the conditions for the whole causal model to be specified, we start with the structure identifiability of partially observed linear causal models. Recent advances have shown that if certain graphical conditions are satisfied [24, 18], even though all variables including latent ones are allowed to be very flexibly related, the causal structure can still be identified. Next, we focus on the conditions by [18], which takes that of [24] as special cases. Roughly speaking, the identifiability of the structure of a partially observed linear causal model is built upon the identifiability of atomic covers, defined as follows (with *effective cardinality* defined as $||\mathcal{V}|| = |(\cup_{\mathbf{V} \in \mathcal{V}} \mathbf{V})|$ and $PCh_\mathcal{G}$ defined in Appendix B.2).

**Definition 4** (Atomic Cover [18]). *Let $\mathbf{V} \in \mathbf{V}_\mathcal{G}$ be a set of variables, where $l$ out of $|\mathbf{V}|$ are latent, and the remaining $|\mathbf{V}| - l$ are observed. $\mathbf{V}$ is an atomic cover if $\mathbf{V}$ is a single observed variable, or if the following conditions hold:*

(i) *There exists a set of atomic covers $\mathcal{C}$, with $||\mathcal{C}|| \geq l + 1$, such that $\cup_{\mathbf{C} \in \mathcal{C}} \mathbf{C} \subseteq PCh_\mathcal{G}(\mathbf{V})$.*

(ii) *There exists a set of covers $\mathcal{N}$ with $||\mathcal{N}|| \geq l + 1$, s.t. $(\cup_{\mathbf{N} \in \mathcal{N}} \mathbf{N}) \cap (\cup_{\mathbf{C} \in \mathcal{C}} \mathbf{C}) = \emptyset$, every element in $\cup_{\mathbf{N} \in \mathcal{N}} \mathbf{N}$ is a neighbour of every element in $\mathbf{V}$, and $\mathbf{V}$ d-separates $\mathcal{N}$ and $\mathcal{C}$.*

(iii) *There does not exist a partition $\mathcal{P}$ of $\mathbf{V}$, s.t., all elements in $\mathcal{P}$ are atomic covers.*

The intuition that we build structure identifiability upon the notion of atomic covers is as follows. When a set of latent variables share the same set of children and neighbors, it is impossible to differentiate these latent variables from each other, and thus we need to consider them together as the minimal identifiable group to build up the identifiability of the whole structure. Such a minimal identifiable group of variables is defined as an atomic cover. Roughly, for a group of variables to be qualified as an atomic cover, it has to have enough children and neighbors. An example is as follows.

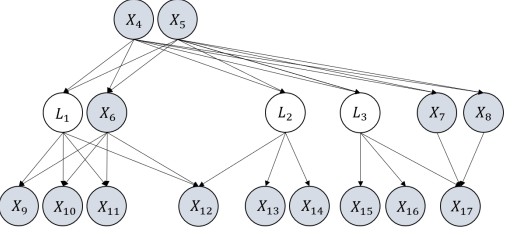

Figure 3: An illustrative graph that satisfies the conditions for structure-identifiability. At the same time, it also satisfies the condition for parameter identifiability - given the structure and $\Sigma_\mathbf{X}$, all the parameters are identifiable only up to group sign indeterminacy.

**Example 2** (Example of Atomic Cover). *Consider the graph in Fig. 3. $\mathbf{V} = \{\mathsf{L}_1, \mathsf{X}_6\}$ is an atomic cover. This is because there exist $\mathcal{C} = \{\{\mathsf{X}_9\}, \{\mathsf{X}_{10}\}\}$ with $||\mathcal{C}|| \geq l + 1 = 2$ such that (i) in Def. 4 is satisfied. And there exist $\mathcal{N} = \{\{\mathsf{X}_{11}\}, \{\mathsf{X}_{12}\}\}$ (or, $\mathcal{N} = \{\{\mathsf{X}_4\}, \{\mathsf{X}_5\}\}$) with $||\mathcal{N}|| \geq l + 1 = 2$ such that (ii) in Def. 4 is satisfied. We can also find that (iii) in Def. 4 is satisfied. Thus $\{\mathsf{L}_1, \mathsf{X}_6\}$ is an atomic cover. Another example would be in Figure 8, where $\{\mathsf{L}_1, \mathsf{L}_2\}$ is an atomic cover.*

**Condition 1** (Basic Conditions for Structure Identifiability [18]). *$\mathcal{G}$ satisfies the basic graphical condition for identifiability, if every latent variable belongs to at least one atomic cover in $\mathcal{G}$ and for each atomic cover with latent variables, any of its children is not adjacent to any of its neighbours.*

**Condition 2** (Condition on Colliders [18]). *In $\mathcal{G}$, if (i) there exists sets of variables $\mathbf{V}$, $\mathbf{V_1}$, $\mathbf{V_2}$, and $\mathbf{T}$ such that every variable in $\mathbf{V}$ is a collider of two atomic covers $\mathbf{V_1}$, $\mathbf{V_2}$, and $\mathbf{T}$ is a minimal set of variables that d-separates $\mathbf{V_1}$ from $\mathbf{V_2}$, and (ii) there exists at least one latent variable in $\mathbf{V} \cup \mathbf{V_1} \cup \mathbf{V_2} \cup \mathbf{T}$, then we must have $|\mathbf{V}| + |\mathbf{T}| \geq |\mathbf{V_1}| + |\mathbf{V_2}|$.*

**Example 3** (Example that satisfies Conditions 1 and 2). *Consider Figure 3. All latent variables in the graph belong to at least one atomic cover and thus Condition 1 is satisfied. Plus, Condition 2 is also satisfied. This is because the sets of variables $\mathbf{V}$, $\mathbf{V_1}$, $\mathbf{V_2}$, and $\mathbf{T}$ that satisfy (i) and (ii) in Condition 2 are $\mathbf{V} = \{X_{12}\}$, $\mathbf{V_1} = \{L_1, X_6\}$, $\mathbf{V_2} = \{L_2\}$, and $\mathbf{T} = \{X_4, X_5\}$, and we also have $|\mathbf{V}| + |\mathbf{T}| \geq |\mathbf{V_1}| + |\mathbf{V_2}|$. Therefore, the graph in Figure 3 satisfies both Conditions 1 and 2.*

The identifiability theory of structure is as follows. For a graph $\mathcal{G}$, if Condition 1 and Condition 2 are satisfied, then asymptotically the structure is identifiable up to the Markov equivalence class (MEC) of $\mathcal{O}_a(\mathcal{O}_s(\mathcal{G}))$ (definitions of $\mathcal{O}_a(\cdot)$ and $\mathcal{O}_s(\cdot)$ can be found in Appendix B.4). Roughly speaking, the underlying causal structure of $\mathcal{G}$ can be identified except that the directions of some edges cannot be determined. Next, we will show that, given any DAG in the identified equivalence class together with $\Sigma_{\mathbf{X}}$, the parameters of the model are also identifiable, if certain conditions are satisfied.

### 3.3 Identifiability of Parameters

In this section we show that, given graphical Conditions 1 and 2, the causal coefficients $F$ in Definition 1 are also identifiable, if certain conditions are satisfied.

**Theorem 3** (Sufficient Condition for Parameter Identifiability (up to group sign), Based on Structure Identifiability). *Assume that $\mathcal{G}$ satisfies Conditions 1 and 2 and thus the structure can be identified up to the MEC of $\mathcal{O}_a(\mathcal{O}_s(\mathcal{G}))$. For any DAG in the equivalence class, the parameters are identifiable, if both the following hold:*

*(i) For any atomic cover $\mathbf{V} = \mathbf{X} \cup \mathbf{L}$, $|\mathbf{L}| \leq 1$.*

*(ii) If an atomic cover $\mathbf{V} = \mathbf{X} \cup \mathbf{L}$ satisfies $|\mathbf{L}| \neq 0$ and $|\mathbf{X}| \geq 1$, then all simple treks (Def. 5) between $\mathbf{L}$ and $\mathbf{X}$ do not contain any latent variables that are not in $\mathbf{L}$.*

Theorem 3 provides a sufficient condition such that the parameters are identifiable. Now, for a better understanding of Theorem 3, we provide an example of it as follows.

**Example 4** (Example for Theorem 3). *The graph $\mathcal{G}$ in Figure 3 satisfies the conditions for parameter identifiability in Theorem 3. Specifically, condition (i) in Theorem 3, is satisfied as all atomic covers contain no more than one latent variable. Plus, condition (ii) in Theorem 3 is also satisfied, as the atomic cover $\mathbf{V} = \mathbf{X} \cup \mathbf{L} = \{L_1\} \cup \{X_6\}$ satisfies $|\mathbf{L}| \neq 0$ and $|\mathbf{X}| \geq 1$ and all simple treks between $\{L_1\}$ and $\{X_6\}$ contain only observed variables except $\{L_1\}$. Therefore, the parameters are identifiable for the graph in Figure 3.*

Next, we discuss under which conditions the parameters are guaranteed to be not identifiable. As discussed in Section 3.1, there are three kinds of indeterminacy. The first one can be solved by assuming unit variance of the noise terms of latent variables while the second one group sign indeterminacy is rather trivial such that we still consider parameters as identifiable even if group sign indeterminacy exists. Therefore, we will focus on the third one, orthogonal transformation indeterminacy, in what follows.

**Theorem 4** (Condition for the Existence of Orthogonal Transformation Indeterminacy). *Consider the model in Definition 1. If a set of latent variables $\bar{\mathbf{L}}$ with $|\bar{\mathbf{L}}| \geq 2$, have the same parents and children, then there must exist orthogonal transformation indeterminacy regarding the edge coefficients $F$. In other words, $F$ can at most be identified up to orthogonal transformation indeterminacy.*

**Example 5** (Example for Thm. 4). *Consider Fig. 8. The graph satisfies the conditions in Thm. 4 as the parents and children of $L_1$ and $L_2$ are exactly the same. Therefore, there must exist orthogonal transformation indeterminacy for the edge coefficients $F$ and thus the parameters are not identifiable.*

The Theorem 4 above indicates that, if there exist two latent variables that share the same parents and children, then the edge parameters can at most be identified up to orthogonal transformation. This directly implies a necessary condition for parameter identifiability as follows.

**Corollary 1** (General Necessary Condition for Parameter Identifiability). *For parameters to be identifiable, every pair of latent variables has to have at least one different parent or child.*

Corollary 1 captures a necessary condition in the general cases such that parameters are identifiable. If we further consider the graphs that are also structure identifiable (as we need to identify the structure first to fully specified the causal model), we further have the following Corollary 2 by considering the notion of atomic covers (the proofs of both corollaries can be found in the Appendix).

**Corollary 2** (Necessary Condition about Atomic Covers for Parameter Identifiability). *Assume $\mathcal{G}$ satisfies Conditions 1 and 2 and thus the structure can be identified up to the MEC of $\mathcal{O}_a(\mathcal{O}_s(\mathcal{G}))$. For any DAG $\mathcal{G}$ in the equivalence class, for $\mathcal{G}$'s parameters to be identifiable, every atomic cover must contain no more than one latent variable.*

**Remark 3** (Necessity of Conditions in Theorem 3). *Condition (i) in Theorem 3 is provably necessary: by Corollary 2, for parameters to be identifiable, one has to assume (i) in Theorem 3.*

Establishing a necessary and sufficient condition is always highly non-trivial in various tasks. For example, for the identification of linear non-Gaussian causal structure with latent variables, researchers initially developed sufficient conditions with three pure children in [46], later relaxed to two in [11, 58], before ultimately achieving both necessary and sufficient conditions in [1]. Similarly, for parameter identification, although the condition we proposed is not a necessary and sufficient one, it could serve as a stepping stone towards tighter and ultimately the necessary and sufficient condition for the field.

Below, we also provide a sufficient condition for parameter identifiability that does not rely on structure identifiability in Theorem 5. It is particularly useful when the structure is directly given by some domain experts.

**Theorem 5** (Sufficient Condition for Parameter Identifiability (up to group sign) without Requiring Structure Identifiability). *In $\mathcal{G}$, if for every latent variable $\mathsf{L}$ there always exist another three distinct variables (which can be latent or observed), such that two of the three are pure children of $\mathsf{L}$ and the rest one is a neighbor of $\mathsf{L}$, then the parameters are identifiable.*

Identifiability theory often focuses on the asymptotic case, i.e., we assume that we know the structure and the population covariance matrix $\Sigma_{\mathbf{X}}$. However, in practice, we only have access to i.i.d. data with finite sample size and thus only have the sample covariance matrix. Therefore, in the next section, we will propose a novel method to estimate the parameters in the finite sample cases.

## 4 Parameter Estimation Method

### 4.1 Objective

Our goal is to estimate $F$ in Definition 1, given the causal structure $\mathcal{G}$ and observational data. The key is to parameterize the population covariance $\Sigma_{\mathbf{X}}$ using $\theta = (F, \Omega)$ and then maximize the likelihood of observed sample covariance $\hat{\Sigma}_{\mathbf{X}}$. To make this technically precise, we provide a closed-form expression of $\Sigma_{\mathbf{X}}$ in terms of $\theta$ in the following proposition, with a proof given in Appendix A.7.

**Proposition 1** (Parameterization of Population Covariance). *Consider the model defined in Def. 1. Let $M := \left(I - F_{\mathbf{LL}} - F_{\mathbf{LX}}(I - F_{\mathbf{XX}})^{-1}F_{\mathbf{XL}}\right)^{-1}$ and $N := \left((I - F_{\mathbf{LL}})F_{\mathbf{XL}}^{-1}(I - F_{\mathbf{XX}}) - F_{\mathbf{LX}}\right)^{-1}$. Then, the population covariance matrices of $\mathbf{L}$ and $\mathbf{X}$ can be formulated as*

$$\Sigma_{\mathbf{L}} = M^T \Omega_{\epsilon_{\mathbf{L}}} M + N^T \Omega_{\epsilon_{\mathbf{X}}} N, \tag{1}$$

$$\Sigma_{\mathbf{X}} = (I - F_{\mathbf{XX}})^{-T}\left(F_{\mathbf{LX}}^T \Sigma_{\mathbf{L}_{\mathcal{G}}} F_{\mathbf{LX}} + \Omega_{\epsilon_{\mathbf{X}}} + \Omega_{\epsilon_{\mathbf{X}}} N F_{\mathbf{LX}} + F_{\mathbf{LX}}^T N^T \Omega_{\epsilon_{\mathbf{X}}}\right)(I - F_{\mathbf{XX}})^{-1}. \tag{2}$$

The formulations of $\Sigma_{\mathbf{L}}$ and $\Sigma_{\mathbf{X}}$ are rather complicated due to the general scenario we considered, i.e., latent variables can be the cause or the effect of latent and observed variables. That is, the submatrices $F_{\mathbf{LL}}, F_{\mathbf{LX}}, F_{\mathbf{XL}}$ and $F_{\mathbf{XX}}$ defined in the above proposition can all have nonzero entries. In most existing works, at least one of these submatrices are assumed to be zero. For instance, factor analysis assumes that $F_{\mathbf{LL}}, F_{\mathbf{XL}}$ and $F_{\mathbf{XX}}$ are zero, while [32] assumes that $F_{\mathbf{LL}}$ and $F_{\mathbf{XL}}$ are zero. Furthermore, Proposition 1 also provides insight into the indeterminacy involved when identifying the parameters, such as the indeterminacy of variance in Theorem 1 and the orthogonal transformation indeterminacy in Theorem 4.

Similar to factor analysis [45, 7, 21], we assume $\epsilon_{\mathbf{V}}$ are Gaussian and thus $\mathbf{X}$ are jointly Gaussian. Thus, the negative log-likelihood of observational data can be formulated as

$$\mathcal{L} = (K/2)(\mathrm{tr}((\Sigma_{\mathbf{X}})^{-1}\hat{\Sigma}_{\mathbf{X}}) + \log \det \Sigma_{\mathbf{X}}), \tag{3}$$

where $K$ is the number of i.i.d. observations. With the parameterized negative log-likelihood, we estimate the edge coefficients by minimizing the negative log-likelihood, as

$$\hat{F}, \hat{\Omega} = \arg\min_{F,\Omega} \mathcal{L}, \quad \text{subject to } \Omega_{\epsilon_{\mathbf{L}}} = I, \tag{4}$$

where the entries of matrix $F$ that do not correspond to an edge in $\mathcal{G}$ are constrained to be zero during the optimization.

Note that in Eq. (4) the constraint that the noise terms of latent variables have unit variance is crucial to deal with the variance indeterminacy defined in Theorem 1. In practice, it is also favorable to use another constraint to address the variance indeterminacy, i.e., the constraint that all the latent variables have unit variance. This leads to an alternative objective as

$$\hat{F}, \hat{\Omega} = \arg\min_{F,\Omega} \mathcal{L}, \quad \text{subject to } (\Sigma_{\mathbf{L}})_{ii} = 1, \ i \in [m], \tag{5}$$

where the entries of $F$ that do not correspond to an edge in $\mathcal{G}$ are also constrained to be zero.

Both objectives in Eqs. (4) and (5) can be employed, and yet using the second one gives rise to edge coefficients that are easier to understand. To be concerete, if we normalize all observed variables to have unit variance, then using Eq. (5) would give rise to $\hat{F}$ such that $-1 \le \hat{F}_{i,j} \le 1, \forall i, j \in [m]$. An example can be found in Figure 4. However, it may not be straightforward to realize the constraint in Eq. (5). To this end, in the next section we introduce a way to parameterize $\Sigma_{\mathbf{X}}$ using $F$, such that the required constraint in Eq. (5) can be automatically satisfied. Later in Section 5.2, we also empirically compare the performance of using Eq. (4) with that of using Eq. (5).

### 4.2 Parameterization Trick of Covariance Matrix

In this section, we introduce how trek rules can be employed to parameterize $\Sigma_{\mathbf{X}}$ while the unit variance constraint on latent variables in Eq. (5) can be elegantly satisfied. We start with the definition of trek. For readers who are less familiar with treks, please refer to Appendix B.1 for examples.

**Definition 5** (Treks [50]). *In $\mathcal{G}$, a trek from $\mathsf{X}$ to $\mathsf{Y}$ is an ordered pair of directed paths $(P_1, P_2)$ where $P_1$ has a sink $\mathsf{X}$, $P_2$ has a sink $\mathsf{Y}$, and both $P_1$ and $P_2$ have the same source $\mathsf{Z}$, i.e., $top(P_1, P_2) = \mathsf{Z}$. A Trek is simple if $P_1$ and $P_2$ have no intersection except their common source $\mathsf{Z}$.*

At this point, we are able to parameterize each entry of $\Sigma_{\mathbf{X}}$ using $(F, \{\sigma_{ii}\}_{i=1}^{n+m})$, instead of $(F, \Omega)$, by making use of the (simple) trek rule [50], as follows:

$$\sigma_{ij} = \sum_{P_1, P_2 \in \mathcal{S}(\mathsf{V}_i, \mathsf{V}_j)} \sigma_{\text{top}(P_1, P_2)} f^{P_1} f^{P_2}, \tag{6}$$

where $\mathcal{S}(\mathsf{V}_i, \mathsf{V}_j)$ is the set of all simple treks between $\mathsf{V}_i$ and $\mathsf{V}_j$, and $f^P$ is the path monomial along $P$ defined as $f^P := \Pi_{k \to l \in P} f_{kl}$.

By this form of parameterization, we can simply set all entries of $\{\sigma_{ii}\}_{i=1}^{n+m}$ as 1 (which is equivalent to requiring all variables to have unit variance), such that the constraint in Eq. (5) can be automatically satisfied. For a better understanding of how to use the simple trek rule for parameterization, we provide an example as follows.

**Example 6** (Example for Parameterization using Simple Trek). *In Figure 7 (a), there are four simple treks between $\mathsf{X}_4$ and $\mathsf{X}_5$, as shown in (b). By the simple trek rule and further assuming that all variables have unit variance, the covariance between $\mathsf{X}_4$ and $\mathsf{X}_5$, $\sigma_{4,5}$, can be formulated as $f_{1,4}f_{1,5} + f_{3,4}f_{3,5} + f_{2,1}f_{1,4}f_{2,3}f_{3,5} + f_{2,3}f_{3,4}f_{2,1}f_{1,5}$.*

## 5 Experiments

We validate our identifiability theory and parameter estimation method on synthetic and real-life data.

### 5.1 Setting and Evaluation Metric

We begin with our experimental setting of synthetic data. The causal strength $f_{ij}$ is uniformly sampled from $[-2, 2]$ and the noise terms are Gaussian with variance uniformly from $[1, 5]$. We consider 20 graphs. 10 of them should be parameter-identifiable up to group sign indeterminacy according to our identifiability theory and we refer to them as *GS Case* (examples in Figure 10 in Appendix). Another 10 should be parameter-identifiable up to group sign and orthogonal transformation indeterminacy and we refer to them as *OT Case* (examples in Figure 11 in Appendix). On average each graph contains 15 variables, 3 out of them are latent. We consider three different sample sizes: 2k, 5k, and 10k. We use three random seeds to generate the causal model and report the mean performance as well as the std.

Table 1: Experimental result on synthetic data using MSE (mean (std)).

| | | MSE up to group sign | | | | MSE up to orthogonal. | |
|---|---|---|---|---|---|---|---|
| Method | | Estimator | Estimator-TR | Method | | Estimator | Estimator-TR |
| GS Case | 2k | 0.0023 (0.002) | 0.0012 (0.0005) | OT Case | 2k | 0.0278 (0.008) | 0.0355 (0.015) |
| | 5k | 0.0014 (0.002) | 0.0005 (0.0005) | | 5k | 0.0194 (0.002) | 0.0352 (0.012) |
| | 10k | 0.0012 (0.001) | 0.0003 (0.0004) | | 10k | 0.0182 (0.003) | 0.0351 (0.015) |

(a) MSE up to group sign indeterminacy.     (b) MSE up to orthogonal transformation.

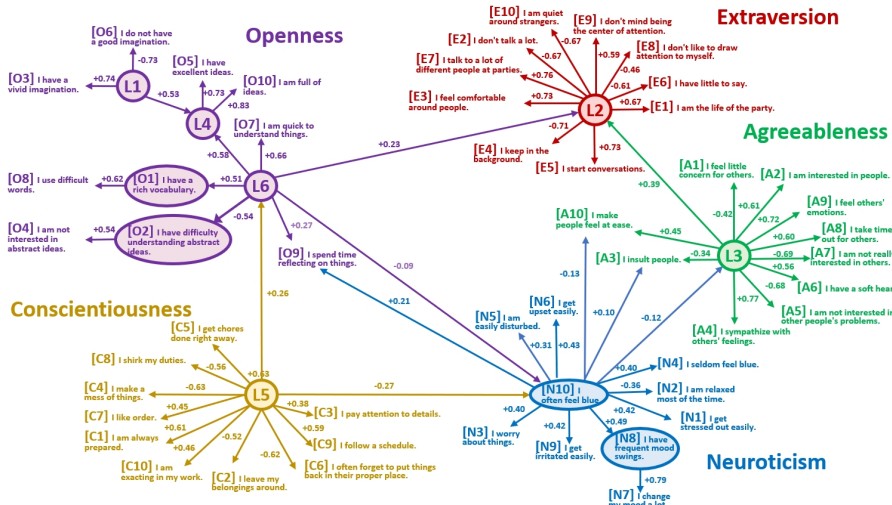

Figure 4: Estimated edge coefficients by the proposed method for Big Five human personality dataset. Variables whose name starts with "L" are latent variables while the others are observed variables.

As the optimization in Eq (4) is nonconvex, we will rely on 30 random starts and choose the one with the best likelihood. We report the performance of the proposed method with two different objectives. **(i)** Parameter Estimator with objective defined in Eq. (4), referred to as Estimator, and **(ii)** Parameter Estimator with objective defined in Eq. (5) and Trek Rule parameterization trick in Eq (6), referred to as Estimator-TR.

It is worth noting that our setting is very general in that we allow latent variables and observed variables to be causally connected in a very flexible way, and we consider the identification of parameters of edges that can involve both observed and latent variables. Therefore, to the best of our knowledge, no current method can achieve the same goal to serve as the baseline (which also shows the novelty of the proposed method). As such, we mainly focus on comparing our estimation result with the ground truth parameters. We use two MSE-based metrics defined as follows.

**MSE up to group sign:** suppose the ground truth parameter is $F$ and our estimation is $\hat{F}$. The MSE up to group sign is defined as $\frac{\||F|-|\hat{F}|\|_2^2}{\|F\|_0}$, where $|\cdot|$ takes element wise absolute value, $\|\cdot\|_2$ denotes the Frobenius norm and $\|\cdot\|_0$ denotes the number of nonzero entries of a matrix.

**MSE up to orthogonal transformation:** the MSE up to orthogonal transformation is defined as

$$\min_{Q:Q^T Q=I} \frac{\||F_{\mathbf{LL}}| - |Q^T \hat{F}_{\mathbf{LL}} Q|\|_2^2 + \||F_{\mathbf{Lx}}| - |Q^T \hat{F}_{\mathbf{Lx}}|\|_2^2 + \||F_{\mathbf{XL}}| - |\hat{F}_{\mathbf{XL}} Q|\|_2^2 + \||F_{\mathbf{XX}}| - |\hat{F}_{\mathbf{XX}}|\|_2^2}{\|F\|_0}, \quad (7)$$

where the optimization is solved by Adam [27] and the orthogonal matrix $Q$ can be directly parameterized in PyTorch.

### 5.2 Synthetic Data Performance

We report the performance using synthetic data in Tables 1a and 1b, where both our Estimator and Estimator-TR achieve very good identification performance. For example, in the GS scenario with 10k samples, our Estimator achieves 0.0.0012 MSE up to group sign and our Estimator-TR achieves 0.0003 MSE up to group sign. The good performance by Estimator and Estimator-TR not only validates our estimation method, but also empirically verifies our identifiability theory.

### 5.3 Misspecification Behavior

In this section, we show that the proposed estimation method still performs well, even under model misspecification: violation of normality and violation of linearity.

As for violation of normality, we use uniform noise terms for the underlying model, and thus the distribution is not jointly Gaussian anymore. We aim to see to what extent can the proposed method still recover the correct parameters. The result is shown in Table 2 in the Appendix, which shows even when the normality is violated, we can still estimate the parameters pretty well. The reason lies in that our proposed asymptotic identifiability result holds true, even when we do not assume Gaussianity; as we only make use of the second-order statistics of the distribution, the additive noise in Definition 1 can follow any other continuous distribution.

To simulate the violation of linearity, we employ the leaky ReLU function during the generation process, as $V_i = \text{LRELU}(\sum_{V_j \in Pa(V_i)} f_{ji} V_j + \epsilon_{V_i})$, $\text{LRELU}(x) = \max(\alpha x, x)$. When $\alpha$ is close to 1, the function is close to a linear one, and when $\alpha$ is close to 0, the model is very nonlinear. The result is shown in Table 3 and we found that our estimation method is quite robust to small violations of linearity. For example, for Estimator-TR in GS case with 10k sample size, if we set $\alpha = 0.8$, we still get a small MSE of 0.001. Even when $\alpha$ decreases to 0.6, the MSE is around 0.005, which is still small. However, when $\alpha$ is decreased to 0.3, the underlying model is considerably nonlinear, and the MSE increases to 0.027.

### 5.4 Implementation Details, Runtime Analysis, and Scalability

Our code is based on Python3.7 and PyTorch [37]. Data is standardized and the optimization in Eqs. (4), (5), and (7) are solved by Adam [27], with a learning rate of 0.02. We conduct all the experiments with single Intel(R) Xeon(R) CPU E5-2470. All experiments can be finished within 2 hours. We note that our method is very computationally efficient. First, the computational cost is almost irrelevant to sample size: we only need to calculate the sample covariance matrix once and cache it for further use during the optimization. Plus, our estimation method can handle a large number of variables. For example, the running time of our method are roughly 10 seconds, 2 minutes, and 10 minutes for 20 variables, 50 variables, and 100 variables respectively. For 300 variables, which is a considerably large number for typical experiments considered in causal discovery papers, the estimation can still be finished within around one hour.

It is also worth noting that model misspecifications do not influence the computation cost of our method. We briefly discuss the efficiency of checking whether conditions in Theorem 3 hold, together with what if conditions do not hold in solving real-life problems in Appendices A.8 and A.9.

### 5.5 Real-World Data Performance

In this section, we employ a famous psychometric dataset - Big Five dataset `https://openpsychometrics.org/`, to validate our method. It consists of 50 indicators and close to 20,000 data points. There are five dimensions: Openness, Conscientiousness, Extraversion, Agreeableness, and Neuroticism (O-C-E-A-N). Each is measured with 10 indicators. Data is standardized.We employ the RLCD method [18] to determine the MEC and GIN [58] to decide the remaining directions. Then we employ the proposed Estimator-TR to estimate all the edge coefficients. The structure satisfies the condition in Theorem 5 so we know that the parameters are identifiable.

The estimated edge coefficients are shown in Figure 4. We found that our estimated coefficients are well aligned with existing psychology studies. For example, according to [16, 17], being successful in exploratory endeavors depends on the stability to pursue them. This is illustrated in our result where L5$\xrightarrow{+0.26}$L6 and L3$\xrightarrow{+0.39}$L2 indicates that Conscientiousness positively influence openness and Agreeableness positively influences Extraversion. Moreover, it has been shown that people are likely to weigh the outcomes of their actions, thus, their level of Conscientiousness coupled with Neuroticism may prohibit them from engaging in risky behaviors (L5$\xrightarrow{-0.27}$N10$\xrightarrow{-0.12}$L3$\xrightarrow{+0.39}$L2) [54]. Such consistency with current psychometric studies again validates the effectiveness of the proposed method in parameter estimation of real-life systems.

## 6 Conclusion

In this paper, we characterize indeterminacy of parameter identification and provide conditions for identifiability. Finally, we propose a novel estimation method and validate it by empirical study.

## 7   Acknowledgement

This material is based upon work supported by NSF Award No. 2229881, AI Institute for Societal Decision Making (AI-SDM), the National Institutes of Health (NIH) under Contract R01HL159805, and grants from Salesforce, Apple Inc., Quris AI, Florin Court Capital, and the MBZUAI-WIS grant.

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

# A Proofs

## A.1 Proof of Theorem 1

**Theorem 1** (Indeterminacy of Scaling of $\Omega_{\epsilon_L}$). *Consider a model that follows Def. 1 with number of latent variables $m \geq 1$ and $\theta = (F_{LL}, F_{LX}, F_{XL}, F_{XX}, \Omega_{\epsilon_L}, \Omega_{\epsilon_X})$. Let $\Lambda$ be any invertible diagonal matrix, and $\tilde{\theta} = (\tilde{F}_{LL}, \tilde{F}_{LX}, \tilde{F}_{XL}, \tilde{F}_{XX}, \tilde{\Omega}_{\epsilon_L}, \tilde{\Omega}_{\epsilon_X})$, where*

$$\tilde{F}_{LL} = \Lambda^{-1}F_{LL}\Lambda, \ \tilde{F}_{LX} = \Lambda^{-1}F_{LX}, \ \tilde{F}_{XL} = F_{XL}\Lambda, \ \tilde{F}_{XX} = F_{XX}, \ \tilde{\Omega}_{\epsilon_L} = \Lambda^2\Omega_{\epsilon_L}, \ \tilde{\Omega}_{\epsilon_X} = \Omega_{\epsilon_X}.$$

*Then, $\tilde{\theta}$ and $\theta$ entail the same observations, i.e., $\tilde{\Sigma}_X = \Sigma_X$. Furthermore, we have $\tilde{\Sigma}_L = \Lambda\Sigma_L\Lambda$.*

*Proof of Theorem 1.* Let $\begin{pmatrix} F_{LL} & F_{LX} \\ F_{XL} & F_{XX} \end{pmatrix} = \begin{pmatrix} A & B \\ C & D \end{pmatrix}$ and $\begin{pmatrix} \tilde{F}_{LL} & \tilde{F}_{LX} \\ \tilde{F}_{XL} & \tilde{F}_{XX} \end{pmatrix} = \begin{pmatrix} \tilde{A} & \tilde{B} \\ \tilde{C} & \tilde{D} \end{pmatrix}$. Let $M$ and $N$ be matrices defined as in Proposition 1, and similarly for $\tilde{M}$ and $\tilde{N}$. We then have

$$
\begin{aligned}
\tilde{M} &= \left(I - \tilde{A} - \tilde{B}(I - \tilde{D})^{-1}\tilde{C}\right)^{-1} \\
&= \left(\Lambda^{-1}\Lambda - \Lambda^{-1}A\Lambda - (\Lambda^{-1}B)(I - D)^{-1}(C\Lambda)\right)^{-1} \\
&= \Lambda^{-1}\left(I - A - B(I - D)^{-1}C\right)^{-1}\Lambda \\
&= \Lambda^{-1}M\Lambda
\end{aligned}
$$

and

$$
\begin{aligned}
\tilde{N} &= \left((I - \tilde{A})\tilde{C}^{-1}(I - \tilde{D}) - \tilde{B}\right)^{-1} \\
&= \left((\Lambda^{-1}\Lambda - \Lambda^{-1}A\Lambda)(C\Lambda)^{-1}(I - D) - \Lambda^{-1}B\right)^{-1} \\
&= \left((I - A)C^{-1}(I - D) - B\right)^{-1}\Lambda \\
&= N\Lambda.
\end{aligned}
$$

By Proposition 1, the latent covariance matrix $\tilde{\Sigma}_L$ after rescaling of the parameters is given by

$$
\begin{aligned}
\tilde{\Sigma}_L &= \tilde{M}^T\tilde{\Omega}_{\epsilon_L}\tilde{M} + \tilde{N}^T\tilde{\Omega}_{\epsilon_X}\tilde{N} \\
&= (\Lambda^T M^T \Lambda^{-T})(\Lambda\Omega_{\epsilon_L}\Lambda)(\Lambda^{-1}M\Lambda) + \Lambda^T N^T \Omega_{\epsilon_X} N\Lambda \\
&= \Lambda(M^T\Omega_{\epsilon_L}M + N^T\Omega_{\epsilon_X}N)\Lambda \\
&= \Lambda\Sigma_L\Lambda.
\end{aligned}
$$

This implies that the variance of each latent variable $L_i$ is scaled by $\Lambda_{ii}^2$. By Proposition 1, the observed covariance matrix $\tilde{\Sigma}_X$ after rescaling of the parameters is given by

$$
\begin{aligned}
\tilde{\Sigma}_X &= (I - \tilde{D})^{-T}\left(\tilde{B}^T\tilde{\Sigma}_L B + \tilde{\Omega}_{\epsilon_X} + \tilde{\Omega}_{\epsilon_X}\tilde{N}\tilde{B} + \tilde{B}^T\tilde{N}^T\tilde{\Omega}_{\epsilon_X}\right)(I - \tilde{D})^{-1} \\
&= (I - D)^{-T}\bigg((\Lambda^{-1}B)^T(\Lambda\Sigma_L\Lambda)(\Lambda^{-1}B) \\
&\qquad\qquad + \Omega_{\epsilon_X} + \Omega_{\epsilon_X}(N\Lambda)(\Lambda^{-1}B) + (\Lambda^{-1}B)^T(N\Lambda)^T\Omega_{\epsilon_X}\bigg)(I - D)^{-1} \\
&= (I - D)^{-T}\left(B^T\Sigma_L B + \Omega_{\epsilon_X} + \Omega_{\epsilon_X}NB + B^T N^T\Omega_{\epsilon_X}\right)(I - D)^{-1} \\
&= \Sigma_X.
\end{aligned}
$$

$\square$

## A.2 Proof of Theorem 2

**Theorem 2** (Group Sign Indeterminacy). *Consider a model that follows Def. 1 with number of latent variables $m \geq 1$, $\theta = (F_{LL}, F_{LX}, F_{XL}, F_{XX}, \Omega_{\epsilon_L}, \Omega_{\epsilon_X})$, and $\Omega_{\epsilon_L} = I$. Let $S$ be a diagonal sign matrix (entries are either $1$ or $-1$), and $\tilde{\theta} = (\tilde{F}_{LL}, \tilde{F}_{LX}, \tilde{F}_{XL}, \tilde{F}_{XX}, \tilde{\Omega}_{\epsilon_L}, \tilde{\Omega}_{\epsilon_X})$, where*

$$\tilde{F}_{LL} = SF_{LL}S, \ \tilde{F}_{LX} = SF_{LX}, \ \tilde{F}_{XL} = F_{XL}S, \ \tilde{F}_{XX} = F_{XX}, \ \tilde{\Omega}_{\epsilon_L} = \Omega_{\epsilon_L} = I, \ \tilde{\Omega}_{\epsilon_X} = \Omega_{\epsilon_X}.$$

*Then, $\tilde{\theta}$ and $\theta$ entail the same observations, i.e., $\tilde{\Sigma}_X = \Sigma_X$, and $(\tilde{\Sigma}_L)_{ii} = (\Sigma_L)_{ii}$, $\forall i \in [m]$.*

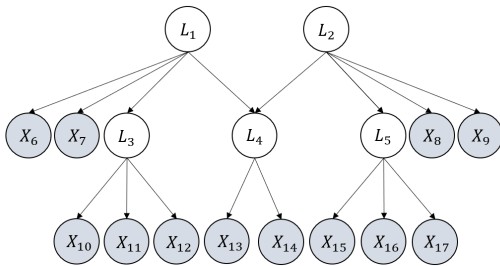

Figure 5: A simple graph that satisfies conditions in Theorem 3, as for each atomic cover with one latent variable, it has no observed variable.

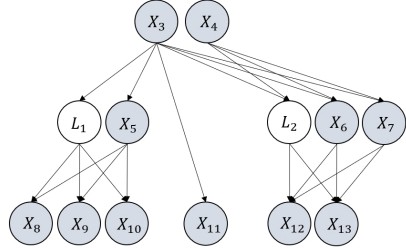

Figure 6: A more complicated graph that satisfies conditions in Theorem 3 and there is an atomic cover that has one latent variable and nonzero observed variables, e.g., $\{L_1, X_5\}$ in the graph. The condition (ii) in Theorem 3 is satisfied in that $L_1, X_5$ can be d-separated by $X_3$.

*Proof of Theorem 2.* Let $\begin{pmatrix} F_{\mathbf{LL}} & F_{\mathbf{LX}} \\ F_{\mathbf{XL}} & F_{\mathbf{XX}} \end{pmatrix} = \begin{pmatrix} A & B \\ C & D \end{pmatrix}$ and $\begin{pmatrix} \tilde{F}_{\mathbf{LL}} & \tilde{F}_{\mathbf{LX}} \\ \tilde{F}_{\mathbf{XL}} & \tilde{F}_{\mathbf{XX}} \end{pmatrix} = \begin{pmatrix} \tilde{A} & \tilde{B} \\ \tilde{C} & \tilde{D} \end{pmatrix}$. Since $S$ is a diagonal sign matrix, we have

$$\tilde{A} := S^{-1}AS, \quad \tilde{B} := S^{-1}B, \quad \tilde{C} := CS, \quad \tilde{D} := D, \quad \tilde{\Omega}_{\epsilon_{\mathbf{L}}} := S^2\Omega_{\epsilon_{\mathbf{L}}}, \quad \text{and} \quad \tilde{\Omega}_{\epsilon_{\mathbf{X}}} := \Omega_{\epsilon_{\mathbf{X}}}.$$

Note that $S$ is an invertible diagonal matrix. By Theorem 1, we have $\tilde{\Sigma}_{\mathbf{X}_{\mathcal{G}}} = \Sigma_{\mathbf{X}_{\mathcal{G}}}$ and $\tilde{\Sigma}_{\mathbf{L}_{\mathcal{G}}} = S\Sigma_{\mathbf{L}_{\mathcal{G}}}S$, and thus $(\tilde{\Sigma}_{\mathbf{L}_{\mathcal{G}}})_{ii} = (\Sigma_{\mathbf{L}_{\mathcal{G}}})_{ii}, \forall i \in [m]$. $\qquad\square$

### A.3  Proof of Theorem 3

The structure identifiability part is that if $\mathcal{G}$ satisfies Condition 1 and Condition 2, the structure of $\mathcal{G}$ can be identified up to the Markov equivalence class of $\mathcal{O}_a(\mathcal{O}_s(\mathcal{G}))$, which is by Theorem 12 in [18].

Next we will focus on the proof of parameter identifiability part, i.e., for any DAG in the equivalence class, if (i) and (ii) in Theorem 3 are satisfied, the parameters are identifiable (up to group sign). Without loss of generality, we assume that all variables have unit variance and zero mean. The reason is that if we can show that the parameters are identfiable (up to group sign) under this assumption, then it is straightforward to show that they are also identifiable under the original assumption where $\Omega_{\epsilon_{\mathbf{L}}} = I$.

**Lemma 1.** *Let $\mathbf{X}, \mathbf{Y}$ be two set of variables, we have $\Sigma_{\mathbf{Y}|\mathbf{X}=x} = \Sigma_{\mathbf{Y}} - \Sigma_{\mathbf{YX}}\Sigma_{\mathbf{X}}^{-1}\Sigma_{\mathbf{XY}}$.*

**Lemma 2.** *Consider a graph $\mathcal{G}$ that satisfies (i) and (ii) in Theorem 3. For an atomic cover $\mathbf{V}$ in $\mathcal{G}$ with one latent variable, $\mathbf{V} = \{L\} \cup \{X_i\}_{i=1}^{k}$ (k could be zero), if it has an observed pure child C and the coefficients of the edges from $\mathbf{V}$ to C, i.e., $f_{L \to C}, f_{X_1 \to C}, \ldots, f_{X_k \to C}$, are known, then for any variable A, such that A $\neq$ C and A is not a descendant of C, $\sigma_{L,A}$ can be calculated as $(\sigma_{A,C} - \Sigma_{i=1}^{k}\sigma_{X_i,A}f_{X_i \to C})/f_{L \to C}$.*

*Proof of Lemma 2.* By the definition of atomic covers, all variables in $\mathbf{V} = \{L\} \cup \{X_i\}_{i=1}^{k}$ are not adjacent. By trek rule and the fact that C is a pure child of $\mathbf{V}$, all treks from C to A go through $\mathbf{V}$, and thus by the trek rule we have $\sigma_{L,A}f_{L \to C} + \Sigma_{i=1}^{k}\sigma_{X_i,A}f_{X_i \to C} = \sigma_{A,C}$. $\qquad\square$

**Remark:** This lemma implies that we can find all edge coefficients of the graph in a bottom-up fashion. Roughly speaking, for a latent variable $L$ that belongs to an atomic cover $\mathbf{V}$, once we identify $f_{L\to C}$, $f_{X_1\to C}$, ..., $f_{X_k\to C}$ where $C$ is an observed pure child of $\mathbf{V}$, we can take $L$ as if it is observed. More specific explanations can be found in the following proof.

**Theorem 3** (Sufficient Condition for Parameter Identifiability (up to group sign), Based on Structure Identifiability). *Assume that $\mathcal{G}$ satisfies Conditions 1 and 2 and thus the structure can be identified up to the MEC of $\mathcal{O}_a(\mathcal{O}_s(\mathcal{G}))$. For any DAG in the equivalence class, the parameters are identifiable, if both the following hold:*

*(i) For any atomic cover $\mathbf{V} = \mathbf{X} \cup \mathbf{L}$, $|\mathbf{L}| \leq 1$.*

*(ii) If an atomic cover $\mathbf{V} = \mathbf{X} \cup \mathbf{L}$ satisfies $|\mathbf{L}| \neq 0$ and $|\mathbf{X}| \geq 1$, then all simple treks (Def. 5) between $\mathbf{L}$ and $\mathbf{X}$ do not contain any latent variables that are not in $\mathbf{L}$.*

*Proof of Theorem 3.* Consider a graph $\mathcal{G}$ that satisfies (i) and (ii) in Theorem 3. We first show that if all the pure children of an atomic cover $\mathbf{V}$ are observed, then all the edge coefficients from the atomic cover to its children are identifiable (up to group sign). To this end, we categorize the scenarios into four cases and prove them separately.

(a) $\mathbf{V} = \{X\}$ contains a single observed variable. The proof for this case is trivial as the edge coefficient from $X$ to its pure child $C$ is simply $\sigma_{X,C}$.

(b) $\mathbf{V} = \{L\}$ contains a single latent variable. By Condition 1 there must exist $C_1$, $C_2$, and $X_N$, such that $C_1$, $C_2$ are pure children of $\mathbf{V}$ and $X_N$ is an observed variable that has a trek to $\mathbf{V}$. Then we have

$$\sigma_{C_1,C_2} = f_{L\to C_1} f_{L\to C_2}, \quad \sigma_{C_1,X_N} = f_{L\to C_1}\sigma_{L,X_N}, \quad \text{and} \quad \sigma_{C_2,X_N} = f_{L\to C_2}\sigma_{L,X_N}.$$

By these three equations, we can solve $f_{L\to C_1}$ and $f_{L\to C_2}$. If $\mathbf{V}$ has more than two pure children, we can prove the identifiability similarly in a pairwise fashion.

(c) $\mathbf{V} = \{L\} \cup \{X_i\}_{i=1}^k$ ($k \geq 1$) contains a single latent variable and $k$ observed variables, where $\mathbf{V}$ has at least three pure children. We assume that there exist $C_1$, $C_2$, and $C_3$, such that $C_1$, $C_2$, and $C_3$ are pure children of $\mathbf{V}$. Let $\sigma_{L|\{X_i\}_{i=1}^k} = t$. In this case, we have

$$\sigma_{C_1,C_2|\{X_i\}_{i=1}^k} = t f_{L\to C_1} f_{L\to C_2}, \tag{8}$$

$$\sigma_{C_1,C_3|\{X_i\}_{i=1}^k} = t f_{L\to C_1} f_{L\to C_3}, \tag{9}$$

$$\sigma_{C_2,C_3|\{X_i\}_{i=1}^k} = t f_{L\to C_2} f_{L\to C_3}. \tag{10}$$

By these three equations, we can solve $f_{L\to C_1}, f_{L\to C_2}$, and $f_{L\to C_3}$, with the only remaining free parameter $t$. In other words, we have $f_{L\to C_1}(t), f_{L\to C_2}(t)$, and $f_{L\to C_3}(t)$.

Next, we show that $\forall i = 1, ..., k, j = 1, ..., 3, f_{X_i\to C_j}$ can be identified. Specifically, as all simple treks between $\mathbf{L}$ and $\mathbf{X}$ contain only observed variables except $\mathbf{L}$, and $\mathbf{L}$ and $\mathbf{X}$ are not directly adjacent, there must exist $\hat{\mathbf{X}}$ such that $\hat{\mathbf{X}}$ d-separates $\mathbf{L}$ and $\mathbf{X}$. Thus, $f_{X_i\to C_j}\sigma_{X_i|\hat{\mathbf{X}}\cup\mathbf{X}\setminus X_i} = \sigma_{X_iC_j|\hat{\mathbf{X}}\cup\mathbf{X}\setminus X_i}$, by which $f_{X_i\to C_j}$ can be solved.

Now we solve $t$. The key is that all the edge coefficients along all simple treks between $\mathbf{X}$ and $\mathbf{L}$ can be identified, by using Lemma 2, with only one free parameter $t$. Thus, we can make use of simple trek rule to parameterize $\Sigma_{L,\mathbf{X}}$ as a function of $t$. By Lemma 1, $\sigma_{L|\mathbf{X}} = t$ can also be formulated as a function of $\Sigma_{L,\mathbf{X}}$, and thus $t$ can be solved. If $\mathbf{V}$ has more than three pure children, we just choose all the combinations of any three pure children.

(d) $\mathbf{V} = \{L\} \cup \{X_i\}_{i=1}^k$ ($k \geq 1$) contains a single latent variable and $k$ observed variables, where $\mathbf{V}$ has two pure children. By Condition 1, there must exist $C_1$, $C_2$ as the pure children of $\mathbf{V}$. If there exists one additional pure child, then it is the same as (c). Plus, if there only exist these two pure children, there must exist $V_N$ such that $V_N$ is a neighbor of $\mathbf{V}$. If $V_N$ is observed, let $X_N = V_N$, otherwise we recursively take $X_N$ as the observed pure children of $V_N$.

Let $\sigma_{L|\{X_i\}_{i=1}^k} = t$. In this case, we have

$$\sigma_{C_1,C_2|\{X_i\}_{i=1}^k} = t f_{L\to C_1} f_{L\to C_2}, \tag{11}$$

$$\sigma_{C_1,X_N|\{X_i\}_{i=1}^k} = t f_{L\to C_1}\sigma_{LX_N}, \tag{12}$$

$$\sigma_{C_2,X_N|\{X_i\}_{i=1}^k} = t f_{L\to C_2}\sigma_{LX_N}. \tag{13}$$

Similar to (c), by solving the above, we have $f_{\mathsf{L}\rightarrow\mathsf{C}_1}(t)$ and $f_{\mathsf{L}\rightarrow\mathsf{C}_2}(t)$.

Next, we show that $\forall i = 1, ..., k, j = 1, ..., 2, f_{\mathsf{X}_i \rightarrow \mathsf{C}_j}$ can be identified. Specifically, as all simple treks between $\mathbf{L}$ and $\mathbf{X}$ contain only observed variables except $\mathbf{L}$, and $\mathbf{L}$ and $\mathbf{X}$ are not directly adjacent, there must exist $\hat{\mathbf{X}}$ such that $\hat{\mathbf{X}}$ d-separates $\mathbf{L}$ and $\mathbf{X}$. Thus, $f_{\mathsf{X}_i \rightarrow \mathsf{C}_j} \sigma_{\mathsf{X}_i | \hat{\mathbf{X}} \cup \mathbf{X} \backslash \mathsf{X}_i} = \sigma_{\mathsf{X}_i \mathsf{C}_j | \hat{\mathbf{X}} \cup \mathbf{X} \backslash \mathsf{X}_i}$, by which $f_{\mathsf{X}_i \rightarrow \mathsf{C}_j}$ can be solved.

Now we solve $t$. The key is that all the edge coefficients along all simple treks between $\mathbf{X}$ and $\mathbf{L}$ can be identified, by using Lemma 2, with only one free parameter $t$. Thus, we can make use of simple trek rule to parameterize $\Sigma_{\mathsf{L},\mathbf{X}}$ as a function of $t$. By Lemma 1, $\sigma_{\mathsf{L}|\mathbf{X}} = t$ can also be formulated as a function of $\Sigma_{\mathsf{L},\mathbf{X}}$, and thus $t$ can be solved.

Taking (a), (b), (c), (d) into consideration, for a graph that satisfies the conditions in Theorem 3, for an atomic cover $\mathbf{V}$ in the graph, if all pure children of it are observed, then all the edge coefficients from $\mathbf{V}$ to its pure children can be identified.

Now, we will prove by induction to show that, for a graph that satisfies the conditions in Theorem 3, for any atomic cover $\mathbf{V}$ in the graph, all the edge coefficients from $\mathbf{V}$ to its children can be identified, and thus all the edge coefficients of the graph can be identified (the set of all edge coefficients in the graph is the union of the set of edge coefficients from each $\mathbf{V}$ to each $\mathbf{V}$'s children).

To this end, we first index all the atomic covers by the inverse causal ordering, such that leaf nodes have smaller indexes. Then we have a sequence of atomic covers $\mathbf{V_i}, i = 1, ..., C$ in the graph, where $C$ is the number of atomic covers in the graph.

(i) We show for $\mathbf{V_i}, i = 1$, all the edge coefficients from $\mathbf{V_1}$ to its children can be identified. This is proved by considering (a) (b) (c) (d), as $\mathbf{V_1}$'s children must be all observed; otherwise it cannot be indexed as 1.

(ii) We show that, for $i > 1$, if for all $\mathbf{V_j}, 1 \le j < i$, all the edge coefficients from $\mathbf{V_j}$ to $\mathbf{V_j}$'s children has been identified, then all the edge coefficients from $\mathbf{V_i}$ to $\mathbf{V_i}$'s children can also be identified. This can be proved by combining (a) (b) (c) (d) with Lemma 2. If $\mathbf{V_i}$ has children that are latent, then the latent children must belong to an atomic cover with a smaller index. Therefore, as all the edge coefficients from $\mathbf{V_j}$ to $\mathbf{V_j}$'s children have been identified, by the use of Lemma 2, the latent children of $\mathbf{V_i}$ can be taken as if they are observed. Therefore, all the edge coefficients from $\mathbf{V_i}$ to $\mathbf{V_i}$'s children can also be identified.

Taking (i) and (ii) together, all the edge coefficients of the graph can be identified.

$\square$

## A.4 Proof of Theorem 5

**Theorem 5** (Sufficient Condition for Parameter Identifiability (up to group sign) without Requiring Structure Identifiability). *In $\mathcal{G}$, if for every latent variable $\mathsf{L}$ there always exist another three distinct variables (which can be latent or observed), such that two of the three are pure children of $\mathsf{L}$ and the rest one is a neighbor of $\mathsf{L}$, then the parameters are identifiable.*

*Proof of Theorem 5.* The proof is a special case of (b) in the proof of Theorem 3. $\square$

## A.5 Proof of Theorem 4

**Lemma 3.** *Let $\Sigma_{\mathbf{X}}$ be the observed covariance matrix entailed by $F_{\mathbf{LL}}, F_{\mathbf{LX}}, F_{\mathbf{XL}}, F_{\mathbf{XX}}, \Omega_{\epsilon_{\mathbf{L}}}, \Omega_{\epsilon_{\mathbf{X}}}$. Let $Q$ be an orthogonal matrix, and*

$$\tilde{F}_{\mathbf{LL}} = Q^T F_{\mathbf{LL}} Q, \ \tilde{F}_{\mathbf{LX}} = Q^T F_{\mathbf{LX}}, \ \tilde{F}_{\mathbf{XL}} = F_{\mathbf{XL}} Q, \ \tilde{F}_{\mathbf{XX}} = F_{\mathbf{XX}}, \ \tilde{\Omega}_{\epsilon_{\mathbf{L}}} = Q^T \Omega_{\epsilon_{\mathbf{L}}} Q, \ and \ \tilde{\Omega}_{\epsilon_{\mathbf{X}}} = \Omega_{\epsilon_{\mathbf{X}}}.$$

*Then, the matrices $\tilde{F}_{\mathbf{LL}}, \tilde{F}_{\mathbf{LX}}, \tilde{F}_{\mathbf{XL}}, \tilde{F}_{\mathbf{XX}}, \tilde{\Omega}_{\epsilon_{\mathbf{L}}}, \tilde{\Omega}_{\epsilon_{\mathbf{X}}}$ can also entail the covariance matrix $\Sigma_{\mathbf{X}}$.*

*Proof of Lemma 3.* Let $\begin{pmatrix} F_{\mathbf{LL}} & F_{\mathbf{LX}} \\ F_{\mathbf{XL}} & F_{\mathbf{XX}} \end{pmatrix} = \begin{pmatrix} A & B \\ C & D \end{pmatrix}$ and $\begin{pmatrix} \tilde{F}_{\mathbf{LL}} & \tilde{F}_{\mathbf{LX}} \\ \tilde{F}_{\mathbf{XL}} & \tilde{F}_{\mathbf{XX}} \end{pmatrix} = \begin{pmatrix} \tilde{A} & \tilde{B} \\ \tilde{C} & \tilde{D} \end{pmatrix}$. Let $M$ and $N$ be matrices defined as in Proposition 1, and similarly for $\tilde{M}$ and $\tilde{N}$. We then have

$$
\begin{aligned}
\tilde{M} &= \left( I - \tilde{A} - \tilde{B}(I - \tilde{D})^{-1}\tilde{C} \right)^{-1} \\
&= \left( Q^T Q - Q^T A Q - (Q^T B)(I - D)^{-1}(CQ) \right)^{-1} \\
&= Q^{-1} \left( I - A - B(I - D)^{-1}C \right)^{-1} Q^{-T} \\
&= Q^T M Q
\end{aligned}
$$

and

$$
\begin{aligned}
\tilde{N} &= \left( (I - \tilde{A})\tilde{C}^{-1}(I - \tilde{D}) - \tilde{B} \right)^{-1} \\
&= \left( (Q^T Q - Q^T A Q)(CQ)^{-1}(I - D) - Q^T B \right)^{-1} \\
&= \left( (I - A)C^{-1}(I - D) - B \right)^{-1} Q^{-T} \\
&= N Q.
\end{aligned}
$$

By Proposition 1, the latent covariance matrix $\tilde{\Sigma}_{\mathbf{L}}$ is given by

$$
\begin{aligned}
\tilde{\Sigma}_{\mathbf{L}} &= \tilde{M}^T \tilde{\Omega}_{\epsilon_{\mathbf{L}}} \tilde{M} + \tilde{N}^T \tilde{\Omega}_{\epsilon_{\mathbf{X}}} \tilde{N} \\
&= (Q^T M^T Q^{-T})(Q^T \Omega_{\epsilon_{\mathbf{L}}} Q)(Q^{-1} M Q) + Q^T N^T \Omega_{\epsilon_{\mathbf{X}}} N Q \\
&= Q^T (M^T \Omega_{\epsilon_{\mathbf{L}}} M + N^T \Omega_{\epsilon_{\mathbf{X}}} N) Q \\
&= Q^T \Sigma_{\mathbf{L}} Q.
\end{aligned}
$$

By Proposition 1, the observed covariance matrix $\tilde{\Sigma}_{\mathbf{X}}$ is given by

$$
\begin{aligned}
\tilde{\Sigma}_{\mathbf{X}} &= (I - \tilde{D})^{-T} \left( \tilde{B}^T \tilde{\Sigma}_{\mathbf{L}} B + \tilde{\Omega}_{\epsilon_{\mathbf{X}}} + \tilde{\Omega}_{\epsilon_{\mathbf{X}}} \tilde{N} \tilde{B} + \tilde{B}^T \tilde{N}^T \tilde{\Omega}_{\epsilon_{\mathbf{X}}} \right)(I - \tilde{D})^{-1} \\
&= (I - D)^{-T} \left( (Q^T B)^T (Q^T \Sigma_{\mathbf{L}} Q)(Q^T B) + \Omega_{\epsilon_{\mathbf{X}}} + \Omega_{\epsilon_{\mathbf{X}}} (NQ)(Q^T B) + (Q^T B)^T (NQ)^T \Omega_{\epsilon_{\mathbf{X}}} \right)(I - D)^{-1} \\
&= (I - D)^{-T} \left( B^T \Sigma_{\mathbf{L}} B + \Omega_{\epsilon_{\mathbf{X}}} + \Omega_{\epsilon_{\mathbf{X}}} N B + B^T N^T \Omega_{\epsilon_{\mathbf{X}}} \right)(I - D)^{-1} \\
&= \Sigma_{\mathbf{X}}.
\end{aligned}
$$

This indicates that the matrices $\tilde{A}, \tilde{B}, \tilde{C}, \tilde{D}, \tilde{\Omega}_{\epsilon_{\mathbf{L}}}$ and $\tilde{\Omega}_{\epsilon_{\mathbf{X}}}$ can also entail the covariance matrix $\Sigma_{\mathbf{X}}$. □

Using Lemma 3, we can prove Theorem 4.

**Theorem 4** (Condition for the Existence of Orthogonal Transformation Indeterminacy). *Consider the model in Definition 1. If a set of latent variables $\mathbf{L}$ with $|\mathbf{L}| \geq 2$, have the same parents and children, then there must exist orthogonal transformation indeterminacy regarding the edge coefficients $F$. In other words, $F$ can at most be identified up to orthogonal transformation indeterminacy.*

*Proof of Theorem 4.* Let $\mathbf{S}_1$, $\mathbf{S}_2$, $\mathbf{S}_3$, $\mathbf{S}_4$, and $\mathbf{S}_5$ be the indices of $\mathbf{L}$, their latent parents, their latent children, their measured parents, and their measured children in $\mathcal{G}$, respectively. Let $Q$ be a $|\mathbf{L}| \times |\mathbf{L}|$ orthogonal matrix. Let $\begin{pmatrix} F_{\mathbf{LL}} & F_{\mathbf{LX}} \\ F_{\mathbf{XL}} & F_{\mathbf{XX}} \end{pmatrix} = \begin{pmatrix} A & B \\ C & D \end{pmatrix}$ and $\begin{pmatrix} \tilde{F}_{\mathbf{LL}} & \tilde{F}_{\mathbf{LX}} \\ \tilde{F}_{\mathbf{XL}} & \tilde{F}_{\mathbf{XX}} \end{pmatrix} = \begin{pmatrix} \tilde{A} & \tilde{B} \\ \tilde{C} & \tilde{D} \end{pmatrix}$.
For matrices $A, B, C$, and $D$ from matrix $F$, suppose that we replace $A_{\mathbf{S}_2, \mathbf{S}_1}$, $A_{\mathbf{S}_1, \mathbf{S}_3}$, $C_{\mathbf{S}_4, \mathbf{S}_1}$, and $B_{\mathbf{S}_1, \mathbf{S}_5}$ with $A_{\mathbf{S}_2, \mathbf{S}_1} Q$, $Q^T A_{\mathbf{S}_1, \mathbf{S}_3}$, $C_{\mathbf{S}_4, \mathbf{S}_1} Q$, and $Q^T B_{\mathbf{S}_1, \mathbf{S}_5}$, respectively. Then, we will show that the entailed covariance matrix $\Sigma_{\mathbf{X}}$ is unchanged.

Let $U$ be an $m \times m$ orthogonal matrix such that: (i) $U_{\mathbf{S}_1, \mathbf{S}_1} = Q$, (ii) the remaining diagonal entries are ones, and (iii) the remaining non-diagonal entries are zeros. Let

$$
\tilde{A} := U^T A U, \quad \tilde{B} := U^T B, \quad \tilde{C} := C U, \quad \tilde{D} := D, \quad \tilde{\Omega}_{\epsilon_{\mathbf{L}}} := U^T \Omega_{\epsilon_{\mathbf{L}}} U = I, \quad \text{and} \quad \tilde{\Omega}_{\epsilon_{\mathbf{X}}} := \Omega_{\epsilon_{\mathbf{X}}}.
$$

By Lemma 3, the matrices above can entail the same covariance matrix $\Sigma_{\mathbf{X}}$.

By construction of $U$, left multiplication of $U^T$ on $B$ only affects $B_{\mathbf{S}_1,*}$; specifically, it is equivalent to replacing $B_{\mathbf{S}_1,*}$ with $Q^T B_{\mathbf{S}_1,*}$. Furthermore, only the columns of $\mathbf{S}_5$ in $B_{\mathbf{S}_1,*}$ will be affected, because those columns correspond to the measured children of $\mathbf{L}$. Therefore, all entries of $\tilde{B}$ are the same as $B$, except that $B_{\mathbf{S}_1,\mathbf{S}_5}$ is replaced with $Q^T B_{\mathbf{S}_1,\mathbf{S}_5}$. Similar reasoning shows that all entries of $\tilde{C}$ are the same as $C$, except that $C_{\mathbf{S}_4,\mathbf{S}_1}$ is replaced with $C_{\mathbf{S}_4,\mathbf{S}_1} Q$.

Now consider $U^T A U$. By the reasoning above, left multiplication of $U^T$ on $A$ only is equivalent to replacing $A_{\mathbf{S}_1,\mathbf{S}_3}$ with $Q^T A_{\mathbf{S}_1,\mathbf{S}_3}$. Further right multiplication of $U$ on $U^T A$ is equivalent to replacing $(U^T A)_{\mathbf{S}_2,\mathbf{S}_1}$ with $(U^T A)_{\mathbf{S}_2,\mathbf{S}_1} Q$. Since $\mathbf{S}_1$, $\mathbf{S}_2$, and $\mathbf{S}_3$ are mutually disjoint, all entries of $\tilde{A} = U^T A U$ are the same as $A$, except that $A_{\mathbf{S}_2,\mathbf{S}_1}$ and $A_{\mathbf{S}_1,\mathbf{S}_3}$ are replaced with $A_{\mathbf{S}_2,\mathbf{S}_1} Q$ and $Q^T A_{\mathbf{S}_1,\mathbf{S}_3}$, respectively.

Hence, for matrices $A, B, C$, and $D$, suppose we replace $A_{\mathbf{S}_2,\mathbf{S}_1}$, $A_{\mathbf{S}_1,\mathbf{S}_3}$, $C_{\mathbf{S}_4,\mathbf{S}_1}$, and $B_{\mathbf{S}_1,\mathbf{S}_5}$ with $A_{\mathbf{S}_2,\mathbf{S}_1} Q$, $Q^T A_{\mathbf{S}_1,\mathbf{S}_3}$, $C_{\mathbf{S}_4,\mathbf{S}_1} Q$, and $Q^T B_{\mathbf{S}_1,\mathbf{S}_5}$, respectively. By the reasoning above, this is equivalent to replacing $A$, $B$, $C$, and $D$ with $\tilde{A}$, $\tilde{B}$, $\tilde{C}$, and $\tilde{D}$, respectively, which (generically) share the same support and entail the same covariance matrix $\Sigma_{\mathbf{X}}$. $\qquad\square$

## A.6 Proof of Corollary 1 and Corollary 2

**Corollary 1** (General Necessary Condition for Parameter Identifiability). *For parameters to be identifiable, every pair of latent variables has to have at least one different parent or child.*

*Proof of Corollary 1.* Proof by contradiction. If it is not the case that every pair of latent variables has to have at least one different parent or child, then there exist $\mathbf{L}$ such that $|\mathbf{L}| \geq 2$ and $\mathbf{L}$ share the same parents and children. Therefore by Theorem 4 there must exist orthogonal transformation indeterminacy regarding $F$, and thus the parameters are not identifiable. $\qquad\square$

**Corollary 2** (Necessary Condition about Atomic Covers for Parameter Identifiability). *Assume $\mathcal{G}$ satisfies Conditions 1 and 2 and thus the structure can be identified up to the MEC of $\mathcal{O}_a(\mathcal{O}_s(\mathcal{G}))$. For any DAG $\mathcal{G}$ in the equivalence class, for $\mathcal{G}$'s parameters to be identifiable, every atomic cover must contain no more than one latent variable.*

*Proof of Corollary 2.* Proof by contradiction. If for a DAG in the equivalence class, there is an atomic cover that has more than one latent variable, then according to the definition of the concerned equivalence class, the latent variables in that atomic cover share the same parents and children. Then by Theorem 4 there must exist orthogonal transformation indeterminacy regarding $F$, and thus the parameters are not identifiable. $\qquad\square$

## A.7 Proof of Proposition 1

**Proposition 1** (Parameterization of Population Covariance). *Consider the model defined in Def. 1. Let* $M := \left(I - F_{\mathbf{LL}} - F_{\mathbf{LX}}(I - F_{\mathbf{XX}})^{-1} F_{\mathbf{XL}}\right)^{-1}$ *and* $N := \left((I - F_{\mathbf{LL}})F_{\mathbf{XL}}^{-1}(I - F_{\mathbf{XX}}) - F_{\mathbf{LX}}\right)^{-1}$. *Then, the population covariance matrices of $\mathbf{L}$ and $\mathbf{X}$ can be formulated as*

$$\Sigma_{\mathbf{L}} = M^T \Omega_{\epsilon_{\mathbf{L}}} M + N^T \Omega_{\epsilon_{\mathbf{X}}} N, \tag{1}$$

$$\Sigma_{\mathbf{X}} = (I - F_{\mathbf{XX}})^{-T} \left( F_{\mathbf{LX}}^T \Sigma_{\mathbf{L}_{\mathcal{G}}} F_{\mathbf{LX}} + \Omega_{\epsilon_{\mathbf{X}}} + \Omega_{\epsilon_{\mathbf{X}}} N F_{\mathbf{LX}} + F_{\mathbf{LX}}^T N^T \Omega_{\epsilon_{\mathbf{X}}} \right)(I - F_{\mathbf{XX}})^{-1}. \tag{2}$$

*Proof of Proposition 1.* Let $F = \begin{pmatrix} F_{\mathbf{LL}} & F_{\mathbf{LX}} \\ F_{\mathbf{XL}} & F_{\mathbf{XX}} \end{pmatrix} = \begin{pmatrix} A & B \\ C & D \end{pmatrix}$.

Since matrices $A$ and $D$ are invertible, using the formula of $2 \times 2$ block matrix inversion [22, Chapter 0.7], we obtain

$$(I - F)^{-1} = \begin{pmatrix} M & -MB(I-D)^{-1} \\ -(I-D)^{-1}CM & (I-D)^{-1} + (I-D)^{-1}CMB(I-D)^{-1} \end{pmatrix},$$

which implies

$$(I - F)^{-T} = \begin{pmatrix} M^T & -M^T C^T (I - D)^{-T} \\ -(I - D)^{-T} B^T M^T & (I - D)^{-T} + (I - D)^{-T} B^T M^T C^T (I - D)^{-T} \end{pmatrix}$$

and

$$(I-F)^{-T}\Omega = \begin{pmatrix} M^T \Omega_{\epsilon_{\mathbf{L}}} & -M^T C^T (I - D)^{-T} \Omega_{\epsilon_{\mathbf{X}}} \\ -(I - D)^{-T} B^T M^T \Omega_{\epsilon_{\mathbf{L}}} & (I - D)^{-T} \Omega_{\epsilon_{\mathbf{X}}} + (I - D)^{-T} B^T M^T C^T (I - D)^{-T} \Omega_{\epsilon_{\mathbf{X}}} \end{pmatrix}.$$

We then have

$$\Sigma_L = M^T \Omega_{\epsilon_{\mathbf{L}}} M + M^T C^T (I - D)^{-T} \Omega_{\epsilon_{\mathbf{X}}} (I - D)^{-1} C M$$
$$= M^T \Omega_{\epsilon_{\mathbf{L}}} M + N^T \Omega_{\epsilon_{\mathbf{X}}} N$$

and

$$\Sigma_X = (I - D)^{-T} B^T M^T \Omega_{\epsilon_{\mathbf{L}}} M B (I - D)^{-1} + (I - D)^{-T} \Omega_{\epsilon_{\mathbf{X}}} (I - D)^{-1}$$
$$+ (I - D)^{-1} \Omega_{\epsilon_{\mathbf{X}}} (I - D)^{-1} C M B (I - D)^{-1} + (I - D)^{-T} B^T M^T C^T (I - D)^{-T} \Omega_{\epsilon_{\mathbf{X}}} (I - D)^{-1}$$
$$+ (I - D)^{-T} B^T M^T C^T (I - D)^{-T} \Omega_{\epsilon_{\mathbf{X}}} (I - D)^{-1} C M B (I - D)^{-1}$$
$$= (I - D)^{-T} \Big( \Omega_{\epsilon_{\mathbf{X}}} + B^T M^T \Omega_{\epsilon_{\mathbf{L}}} M B + \Omega_{\epsilon_{\mathbf{X}}} (I - D)^{-1} C M B + B^T M^T C^T (I - D)^{-T} \Omega_{\epsilon_{\mathbf{X}}}$$
$$+ B^T M^T C^T (I - D)^{-T} \Omega_{\epsilon_{\mathbf{X}}} (I - D)^{-1} C M B \Big) (I - D)^{-1}$$
$$= (I - D)^{-T} \Big( \Omega_{\epsilon_{\mathbf{X}}} + B^T \Sigma_{\mathbf{L}} B + \Omega_{\epsilon_{\mathbf{X}}} N B + B^T N^T \Omega_{\epsilon_{\mathbf{X}}} \Big) (I - D)^{-1}.$$

$\square$

We now discuss how $M$ and $N$ defined in Proposition 1 are invertible. Note that matrices $I - D$ and $I - F$ are invertible because structure $\mathcal{G}$ is acyclic. This implies $\det(I - F) \neq 0$ and $\det(I - D) \neq 0$. Define

$$U = \begin{pmatrix} I & 0 \\ -(I - D)^{-1} C & I \end{pmatrix},$$

which implies

$$(I - F)U = \begin{pmatrix} M & B \\ 0 & I - D \end{pmatrix}$$

and thus

$$\det((I - F)U) = \det(M) \det(I - D).$$

Since $\det(U) = 1$ and $\det(I - F) \neq 0$, we have

$$\det((I - F)U) = \det(I - F) \det(U) \neq 0.$$

By the statement above and $\det(I - D) \neq 0$, we have

$$\det(M) = \frac{\det((I - F)U)}{\det(I - D)} \neq 0,$$

which implies that $M$ is invertible. Similar reasoning can be used to show that $N$ is invertible.

## A.8 Computational Cost of Checking Whether the Conditions in Theorem 3 Hold

Here we want to investigate, given a structure, can we efficiently check whether the proposed sufficient conditions hold? To this end, we generate random graphs and each graph has 100 variables. According to our empirical result, such a check can be done very efficiently. Specifically, on average, given a structure with 100 variables, it only takes our Python code around 3 seconds to check whether the conditions hold.

## A.9 In Practice, What If the Conditions Do not Hold?

Our condition is useful in solving real-life problems. For example, in the psychometric study, we can properly design the questions with domain knowledge following the condition in Theorem 3 such that each single latent variable has enough observed variables as pure children and thus it can be ensured that all parameters are identifiable (as illustrated in our real-life data result in Figure 4).

On the other hand, even though sometimes the questionnaires and data were designed not so well such that the conditions are not satisfied for the identification of parameters, our Theorem 3 is still useful. In this case, we can still make use of our conditions to check the given structure, and find some local sub-structures where our conditions are satisfied. Consequently, it can be ensured that all the parameters of some sub-structures are identifiable, and we can employ our estimation method to find all the edge coefficients of these sub-structures.

# B Additional Definitions, Graphs, Results, and Examples

## B.1 Example of Treks

**Example 7** (Example of Treks). *In Figure 7 (a), there are four treks between $X_4$ and $X_5$: (i) $X_4 \leftarrow L_1 \rightarrow X_5$, (ii) $X_4 \leftarrow X_3 \rightarrow X_5$, (iii) $X_4 \leftarrow L_1 \leftarrow X_2 \rightarrow X_3 \rightarrow X_5$, and (iv) $X_4 \leftarrow X_3 \leftarrow X_2 \rightarrow L_1 \rightarrow X_5$, illustrated in Figure 7 (b).*

## B.2 Definition of Pure Children

**Definition 6** (Parents, Children, and Descendants of a Set of Nodes [18]). *For a set of nodes $\mathbf{X}$ in $\mathcal{G}$, we have $Ch_{\mathcal{G}}(\mathbf{X}) = \cup_{X \in \mathbf{X}} Ch_{\mathcal{G}}(X)$, $Pa_{\mathcal{G}}(\mathbf{X}) = \cup_{X \in \mathbf{X}} Pa_{\mathcal{G}}(X)$, and $De_{\mathcal{G}}(\mathbf{X}) = \cup_{X \in \mathbf{X}} De_{\mathcal{G}}(X)$.*

**Definition 7** (Pure Children of a Set of Nodes [18]). *$\mathbf{Y}$ are pure children of a set of nodes $\mathbf{X}$ in graph $\mathcal{G}$, i.e., $\mathbf{Y} \in PCh_{\mathcal{G}}(\mathbf{X})$, iff all of the following hold: (i) $\mathbf{X} \cap \mathbf{Y} = \emptyset$, (ii) $\mathbf{Y} \subseteq Ch_{\mathcal{G}}(\mathbf{X})$, (iii) $Pa_{\mathcal{G}}(\mathbf{Y}) = \mathbf{X}$, and (iv) $De_{\mathcal{G}}(\mathbf{Y}) \cap \mathbf{X} = \emptyset$.*

## B.3 Definition of Neighbor and MEC

**Definition 8** (Neighbor). *In $\mathcal{G}$, nodes $X$ and $Y$ are neighbor of each other iff there exist an edge from $X$ to $Y$ or an edge from $Y$ to $X$.*

**Definition 9** (Markov Equivalence Class (MEC)). *Two DAGS $\mathcal{G}_1$ and $\mathcal{G}_2$ belong to the same MEC, iif they share the same skeleton and v-structures.*

## B.4 Definition of Rank-invariant Graph Operator

The definitions are as follows with examples.

**Definition 10** (Skeleton Operator [18]). *Given an atomic cover $\mathbf{V}$ in a graph $\mathcal{G}$, and let $\mathcal{S}$ be the set of all atomic covers in $\mathcal{G}$ such that for all $\mathbf{S} \in \mathcal{S}$, $\mathbf{S} \subset \mathbf{V}$. For all $V_1 \in \mathbf{V}$ and all $V_2 \in PCh_{\mathcal{G}}(\mathbf{V}) \backslash (\bigcup_{\mathbf{S} \in \mathcal{S}} PCh_{\mathcal{G}}(\mathbf{S}))$, if $V_1$ and $V_2$ are not adjacent, draw an edge from $V_1$ to $V_2$. We denote such an operator as skeleton operator $\mathcal{O}_s(\mathcal{G})$.*

**Definition 11** (Intra atomic operator). *For every atomic cover $\mathbf{V}$ in structure $\mathcal{G}$, if $|\mathbf{V}| \geq 2$, we add edges between elements in $\mathbf{V}$ such that edges among $\mathbf{V}$ form a fully connected DAG. We denote such an operator as intra atomic operator $\mathcal{O}_a(\mathcal{G})$.*

**Example 8.** *Suppose the original graph is in Figure 9 (a). After the skeleton operator, we have $\mathcal{O}_s(\mathcal{G})$, which is shown in Figure 9 (b). After the intro atomic operator, we have $\mathcal{O}_a(\mathcal{O}_s(\mathcal{G}))$, which is shown in Figure 9 (c).*

## B.5 Graphs for Synthetic Data Experiments

Please refer to Figures 10 and 11.

## B.6 Additional Result under Model Misspecification

Please refer to Tables 2and 3.

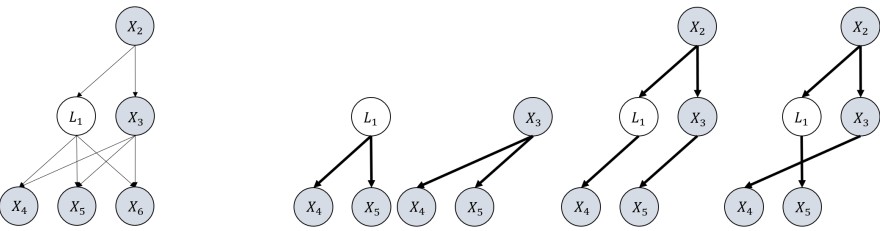

(a) Graph $\mathcal{G}$ to show treks.    (b) The four simple treks between $X_4$ and $X_5$ in (a).

Figure 7: Illustrative figure to show how to parameterize $\Sigma_{\mathbf{X}_{\mathcal{G}}}$ by the use of simple trek rule.

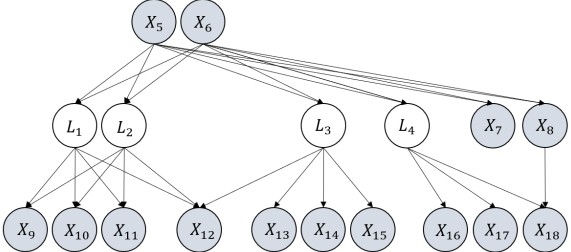

Figure 8: An illustrative graph to show orthogonal transformation indeterminacy. An atomic cover of it, $\{L_1, L_2\}$, has more than one latent variable, and thus there exists orthogonal transformation indeterminacy regarding coefficients of edges that involve $\{L_1, L_2\}$.

## C    Other Discussions

Our optimization problem in Eq. (4) is solved by gradient descent using PyTorch. Our current implementation is based on CPU but it can be further accelerated by using GPU. A very related discussion can also be found in [35].

The optimization problem in Eq. (5) is solved by gradient descent, which involves evaluating the LogDet and matrix inverse (for the gradient) terms (which is similar to continuous causal discovery methods based on Gaussian likelihood [35]). According to [53], the computational complexity is $O(td^3)$, where $d$ is the number of variables and $t$ is the number of iterations of gradient descent respectively. Note that the computational cost is largely independent of the sample size as we only need to calculate the sample covariance once and save it for further use.

It is possible to perform inference on the learned parameters in our framework. To be specific, as we use maximum likelihood estimation for the parameters, some standard techniques can be readily used. For example, bootstrap can be employed to provide standard errors on linear coefficients and Chi-square test can also be done to examine the fitness of the model.

## D    Extended Related Work

One main line of research in latent variable estimation centers on factor-analysis-based methods. Representative studies include [41, 45, 36, 56, 7, 57]. Various other techniques have also been employed for latent structure and parameter identification, including over-complete ICA-based techniques [23, 43, 1] that leverage non-Gaussianity and matrix decomposition-based approaches [3]. However, these approaches typically consider latent variables with observed children, without considering parameter identification in latent hierarchical structures. A more related work is [4], but it considers a much simpler structure.

Another direction would be to project the graph to an ADMG and the latent confounding effects are encoded by correlated noise terms. Following this idea, graphical criteria such as half-trek [20, 5], G-criterion [9], and some further developments [51, 29, 42, 8, 19] has been proposed. Furthermore, another line of works involve studies on causal effect estimation in the presence of latent confounders [52, 6, 30, 33, 26], which often rely on instrumental variables or proxy variables for identification. Notice that in this task, the parameters may not be identified [30], although the causal effect from the treatment variable to the outcome variable can be identified.

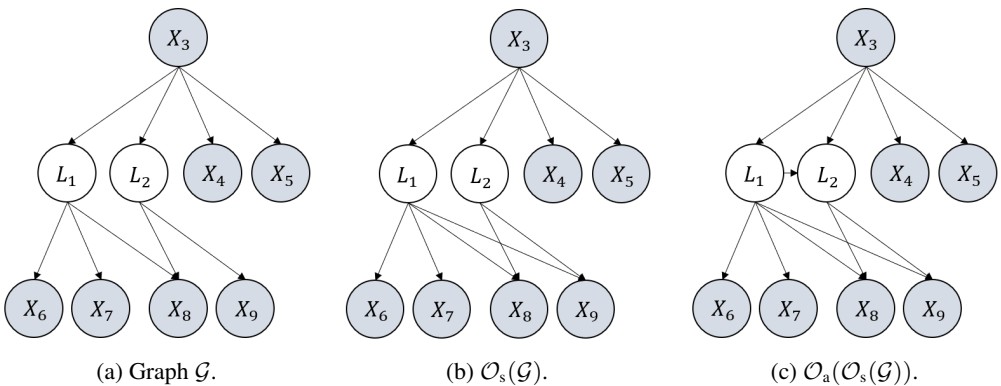

(a) Graph $\mathcal{G}$.          (b) $\mathcal{O}_s(\mathcal{G})$.          (c) $\mathcal{O}_a(\mathcal{O}_s(\mathcal{G}))$.

Figure 9: Example to illustrate graph operators $\mathcal{O}_a$ and $\mathcal{O}_s$.

Table 2: Performance under violation of normality using uniform noise terms in MSE (mean (std)).

| Metric | | MSE up to group sign | |
|---|---|---|---|
| Method | | Estimator | Estimator-TR |
| | 2k | 0.0017 | 0.0005 |
| GS Case | 5k | 0.0018 | 0.0004 |
| | 10k | 0.0018 | 0.0003 |

Furthermore, several existing works also solve an optimization problem that involves parameterization of maximum likelihood, such as those in continuous optimization for causal discovery [35, 59, 31, 10, 34] and parameter estimation of Lyapunov models [55, 15]. Differently, our formulation involving likelihood parameterization aims to estimate parameters of partially observed linear causal models.

# E   Limitations

One limitation of this work is that our theoretical results are based on the assumption of linear gaussian causal models. When data is not linear gaussian, we have also conducted experiments to see the performance of our method. It turns out that our method still performs well in the presence of certain extents of violation of normality and linearity. However, theoretical analysis under violation of linearity and normality would be interesting and the focus of future work.

# F   Broader Impacts

The goal of this paper is to advance the field of machine learning. We do not see any potential negative societal impacts of the work.

Table 3: Performance under violation of linearity using leaky relu in MSE (mean (std)).

| Metric | | MSE up to group sign | |
|---|---|---|---|
| Method | | Estimator | Estimator-TR |
| GS Case with 10k sample size | $\alpha = 0.8$ (close to linear) | 0.004 | 0.001 |
| | $\alpha = 0.6$ (quite nonlinear) | 0.013 | 0.005 |
| | $\alpha = 0.3$ (very nonlinear) | 0.046 | 0.027 |

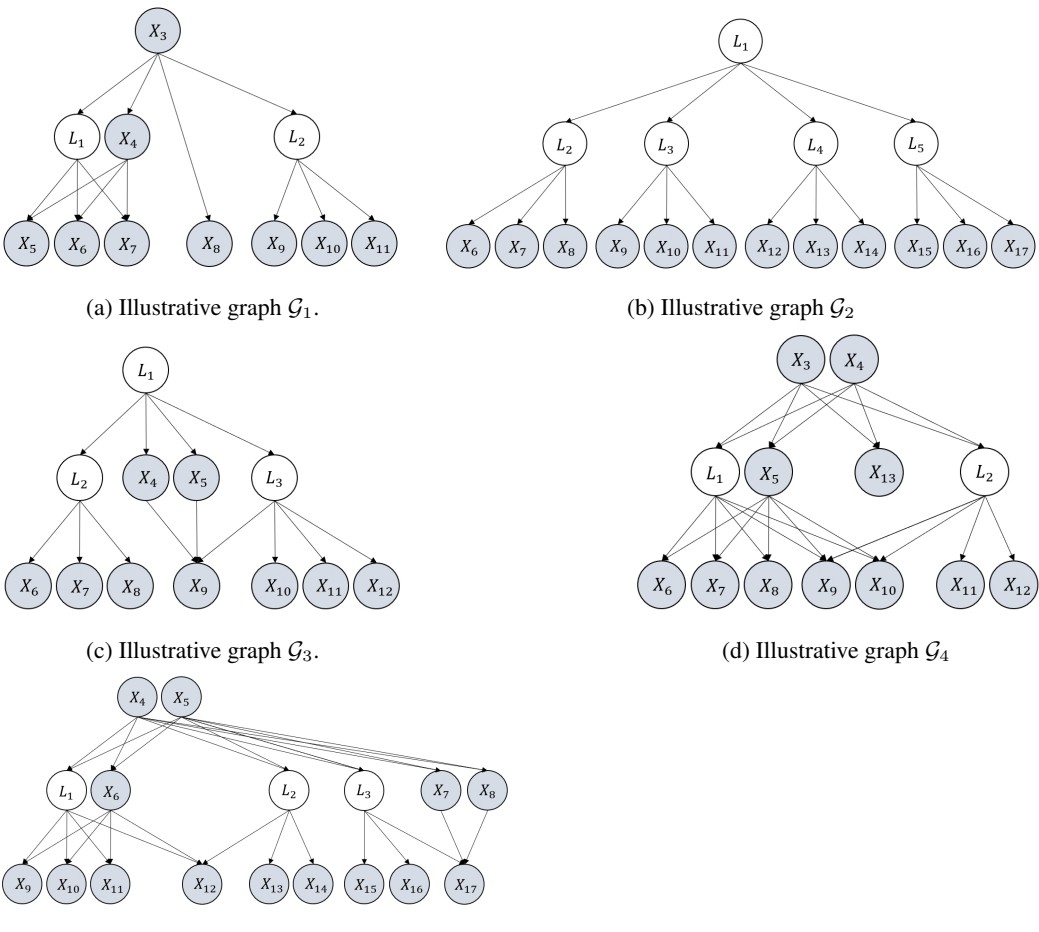

(a) Illustrative graph $\mathcal{G}_1$.

(b) Illustrative graph $\mathcal{G}_2$

(c) Illustrative graph $\mathcal{G}_3$.

(d) Illustrative graph $\mathcal{G}_4$

(e) Illustrative graph $\mathcal{G}_5$

Figure 10: Examples of graphs in the GS case. The parameters of them are identifiable up to group sign indeterminacy.

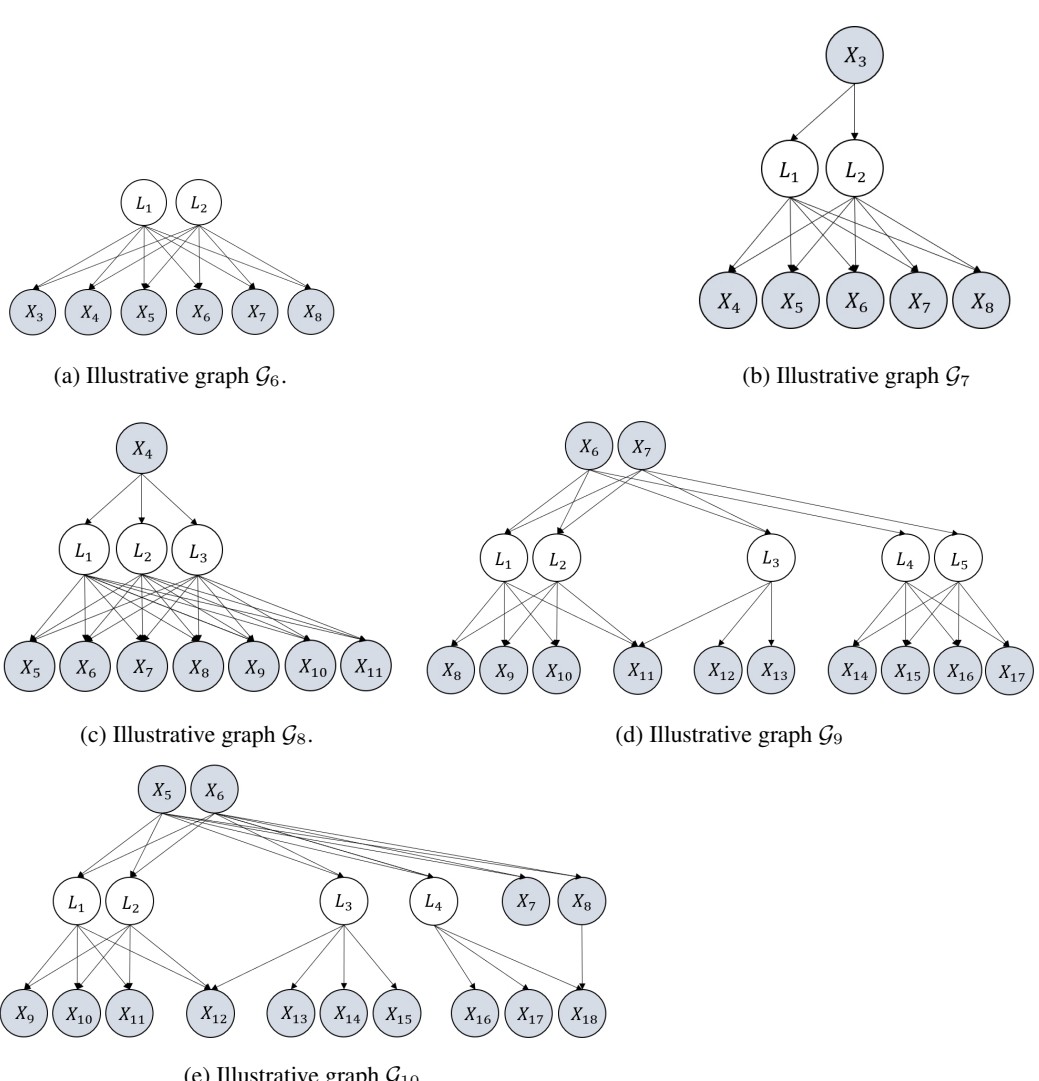

(a) Illustrative graph $\mathcal{G}_6$.

(b) Illustrative graph $\mathcal{G}_7$

(c) Illustrative graph $\mathcal{G}_8$.

(d) Illustrative graph $\mathcal{G}_9$

(e) Illustrative graph $\mathcal{G}_{10}$

Figure 11: Examples of graphs in the OT case. Parameters of them are Identifiable up to group sign and orthogonal transformation indeterminacy.

