# OpenReview forum: "On the Parameter Identifiability of Partially Observed Linear Causal Models"
_NeurIPS.cc/2024/Conference — NeurIPS 2024 poster_

### Official Review · Reviewer_kYam · 2024-06-24

**Soundness:** 4
**Presentation:** 3
**Contribution:** 4
**Rating:** 7
**Confidence:** 4

**Summary:**

The manuscript proposes novel methods for learning linear structural equation models from partially observed data (i.e., allowing for latent variables). The authors provide graphical identifiability conditions for such models and describe an algorithm for learning structural parameters from data via gradient descent. Results on synthetic and real-world examples suggest that the method works well in practice.

**Strengths:**

The manuscript is clear and well-written. It makes a meaningful contribution to a well-studied topic, which I think will be of great interest to many NeurIPS readers and the causality community more generally. The theoretical sections on identifiability and indeterminacy are well reasoned, with helpful examples along the way. The implementation is elegant and efficient, with compelling results on simulated and real-world data.

**Weaknesses:**

I was a bit confused by the discussion of necessary graphical conditions following Thm. 1. First, I would recommend avoiding terms like "pretty close to...necessary", which doesn't mean much. We get a bit more insight in Remark 1, where it's revealed that condition (i) *is* in fact necessary, but condition (ii) is not. We learn that if (ii) does not hold, then "there are...some rare cases where parameters can be identified." Oddly, we don't see any examples of such structures (either in the main text or the supplement), or learn what "rare" amounts to here (presumably *not* Lebesgue measure zero?) Specific counterexamples would help a great deal! Perhaps some sort of disjunctive graphical condition could do the trick, e.g. condition (i) + [(ii) OR (iii)], where (iii) covers those purportedly "rare" structures that violate (ii) but are still technically identifiable. Alternatively, I would cut all discussion of necessity from this section and move it instead to a discussion section later in the manuscript. (I appreciate that space is tight here, but with an extra page in a final version this could be a more satisfying solution).


Minor errata:
-All instances of "upto" should be "up to"
-It appears that references 46 and 47 are the same
-It appears that references 5 and 19 are the same
-The word "Gaussian" is occasionally uncapitalized

**Questions:**

-I take it that the identiability conditions of Thm. 1 are untestable? Or perhaps they have some testable consequences in certain settings? If so, this would be very helpful for practitioners!

-Would it be possible to perform inference on the learned parameters? For instance, could this method provide standard errors on linear coefficients?

-Though I know the method is designed for the partially observable setting, I'm curious how it fares against competitors such as GES or PC when latent variables are absent?

**Limitations:**

Yes, in Appx. D. May be worth moving this to the main text in the final submission, however.

---

> ### Author Rebuttal · Authors · 2024-08-07
>
> Thank you for the time dedicated to reviewing our paper, the insightful comments, and valuable feedback. Please see our point-by-point responses below.
>
> **Q1:** Regarding writing suggestions about Thm 1, Remark 1, and other related discussions.
>
> **A1:**
> Theorem 1 provides a sufficient condition (which consists of conditions (i) + (ii)) for parameters to be identifiable. In Remark 1, we have shown that condition (i) is provably necessary.  Yet, (i)+(ii) is not a necessary and sufficient one, as there exist some cases that fall outside of condition (ii) yet remain identifiable (and they are not of Lebesgue measure zero).
> We genuinely appreciate the reviewer's valuable suggestion and have revised to add multiple examples to illustrate these cases, as you suggested.
> As for disjunctive graphical conditions,
> we totally agree that it could be helpful, e.g., adding a condition (iii) to capture these rare cases such that (i)+(iii) could also be sufficient for identifiability.
> However, characterizing these cases is highly challenging
>  as it may involve tools from algebraic statistics.
>  Thus, we formulate our theorem 1 in the current form and plan to address this gap in future research.
> Regarding other writing suggestions such as moving the discussion of necessity to a section later and moving Appx. D to the main text, and correction of some typos, we
> genuinely thank the reviewer for the valuable feedback and have revised them accordingly.
>
> **Q2:** Test of identifiability conditions will be helpful.
>
> **A2:** Yes, this would indeed be very helpful for practitioners. In this work, we focus on the problem of parameter identification, where the structure is assumed to be given. Thus, testing the conditions in Thm.1 is straightforward in our context.
>
> On the other hand, we believe your question pertains more to structure identification, e.g., whether Conditions 1 and 2 in [17] can be tested.
>  In practical applications, certain strategies can be employed to gain a preliminary understanding of whether the graphical conditions hold; one effective approach is to utilize domain knowledge. Yet, a rigorous test of these graphical assumptions still remains a significant challenge in the field of causal structure learning in general and will be a key area for future research. In this regard, researchers also focus on another critical question: identifying testable consequences when certain conditions are violated, as you mentioned. Specifically for Conditions 1 and 2, if certain kinds of violations exist, consequences can be detected,
> and a detailed discussion can be found in [17]. We thank the reviewer for the valuable question and have added this discussion to our revision.
>
> **Q3:** Would it be possible to perform inference on the learned parameters? For instance, standard errors on linear coefficients?
>
> **A3:**  Yes, it is possible to perform inference in our framework. To be specific, as we use maximum likelihood estimation for the parameters, some standard techniques can be readily used. For example, bootstrap can be employed to provide standard errors on linear coefficients and Chi-square test can also be done to examine the fitness of the model. Thank you for the valuable suggestion and we have added this discussion to our revision.
>
> **Q4:** How it fares against competitors such as GES or PC when latent variables are absent?
>
> **A4:** Yes, it is possible to compare our method with GES - although GES is primarily used for structure identification, it estimates all the linear edge coefficients during the calculation of likelihood. In contrast, constraint-based methods like PC cannot be compared.
> Asymptotically speaking, when latent variables are absent, the parameters estimated by our method will be exactly the same as that of GES, as in this case only the set of true parameters can maximize the likelihood. We also conduct additional experiments to check finite sample cases (5k and 10k data points are concerned) and the result is aligned with the asymptotic analysis - for graphs that have no latent variables, the parameters estimated by our method are nearly the same as that of GES (due to minor difference of implementation details of likelihood).
>
> We genuinely appreciate the reviewer's effort and hope that your concerns/questions are addressed.

---

> > ### Comment · Reviewer_kYam · 2024-08-12
> > **Re: Author rebuttal**
> >
> > Many thanks to the authors for their thoughtful replies to my comments. I will maintain my score and look forward to seeing the revised manuscript in the camera ready draft.

---

> > > ### Author Response · Authors · 2024-08-13
> > >
> > > We would like to once again express our appreciation for your insightful comments, helpful writing suggestions, and positive feedback. If you have any further insights or questions, we would be more than happy to hear from you.
> > >
> > > Many thanks!
> > >
> > > Your sincerely,
> > >
> > > Authors of submission 11557

---

### Official Review · Reviewer_PaWo · 2024-07-08

**Soundness:** 3
**Presentation:** 3
**Contribution:** 3
**Rating:** 5
**Confidence:** 4

**Summary:**

This paper investigates the problem of parameter identification in linear causal models, which is important and well-studied task in causality. The authors examine models that explicitly include both observed and latent variables. The identification of parameters in such models has not been studied so far and the authors are the first to formulate this problem and provide the first results in this regard. The main achievements of the paper are the sufficient and necessary conditions for parameter identifiability.  Moreover the authors conducted empirical studies on both synthetic and real-world data to validate the proposed methods.

**Strengths:**

Thought causal structure learning in the presence of latent variables has been well studied in the literature, the parameter identification in models that explicitly include both observed and latent  variables -- as presented in the submission -- is new. The paper provides non-trivial sufficient and necessary graphical conditions for parameter identifiability and present them in the context of known graphical conditions for structure identifiability provided in [17].

**Weaknesses:**

The sufficient condition for parameter identifiability assumes that G, in addition to conditions (i) and (ii) in Thm. 1, satisfies conditions 1 and 2 presented in section 3.2. I agree that conditions 1 and 2 imply that the structure G can be identified (as shown in [17]) however the  sufficient conditions formulated in this way are overall very restrictive. It would be interesting to have sufficient conditions also in the case of structures which do not satisfy conditions 1 and 2: It can happen that G can be identified even if it does not satisfy 1 and 2 or the structure is provided by a researcher / theory. The authors do not discuss if there exist cases which can be identified but which do not satisfy 1 or 2.

Also, it is not clear what is the gap between the sufficient and necessary conditions.

The next issue is that the authors do not discuss what is the computational complexity of parameter identification in linear causal models, that explicitly include both observed and latent variables.

It is not clear to what extent the sufficient and necessary conditions proposed in Section 3 are useful for (numerical) parameter estimation discussed in Section 4.

**Questions:**

Please provide the motivation for considering edges / parameters from observed to latent variables and edges between latent variables.

See also my questions above.

In Line 69: explain that d=n+m

**Limitations:**

yes

---

> ### Author Rebuttal · Authors · 2024-08-07
>
> Thank you for the time dedicated to reviewing our paper, the insightful comments, and valuable feedback. Please see our point-by-point responses below.
>
> **Q1:** It would be interesting to have sufficient conditions also in structures that do not satisfy conditions 1 and 2: G may be identified even if it does not satisfy 1 and 2 or the structure is provided by a researcher.
>
> **A1:** Thank you for the insightful question. We aim to explore conditions such that the whole causal model can be fully specified from observational data, and thus Conditions 1 and 2 have to be considered for structure identifiability. These conditions are not overly restrictive; rather, they are currently the most general or less restrictive ones for structure identifiability involving latent variables in the linear Gaussian setting, to our best knowledge [17].
>
> At the same time, we also agree that in some cases the structure might be directly given by domain knowledge or experts, and the parameter identifiability for these cases can be interesting. We genuinely thank the reviewer for the suggestion and have added a related discussion to our revision, which can be summarized as follows. If the structure identifiability is not a concern (e.g., when G is given by an expert), a weaker sufficient condition for parameter identifiability can be used: Condition 1, along with conditions (i) and (ii) in Thm 1, are sufficient for parameter identifiability, and Condition 2 is not required.
>
> **Q2:** The gap between the sufficient and necessary conditions?
>
> **A2:** Theorem 1 provides a sufficient condition for parameters to be identifiable. The gap between this sufficient condition and a necessary and sufficient condition is further analyzed in Remark 1. Specifically, the sufficient condition in Thm 1 consists of two parts: condition (i) and condition (ii). In our paper, we have shown that condition (i) is provably necessary, meaning the gap arises only from cases that fall outside of condition (ii) yet remain identifiable. Characterizing these cases is highly challenging as it may involve tools from algebraic statistics, and thus we formulate our theorem 1 in the current form.
>
> We appreciate the reviewer's insightful question and have revised the paper to include illustrative examples of these cases. We plan to address this gap in future research, and hope that these examples inspire further exploration in this area. After all, establishing a necessary and sufficient condition is always highly non-trivial and often requires significant time and multiple efforts (e.g., it takes around 10 years for the structure identification of latent linear non-Gaussian models).
>
> **Q3:**  Computational complexity of parameter identification.
>
> **A3:** The optimization in Eq.3 is solved by gradient descent, which involves evaluating the LogDet and matrix inverse (for the gradient) terms (similar to continuous causal discovery methods based on Gaussian likelihood [33]). According to [58], the computational complexity is $O(td^3)$, where $d$ is the number of variables and $t$ is the number of iterations of gradient descent. Note that the computational cost is largely independent of sample size as we only need to calculate the sample covariance once and save it for further use. Thank you for the valuable suggestion and we have added this analysis on complexity to our revision. As for the empirical runtime analysis, please kindly refer to sec 5.4, where our method is shown to be very efficient - e.g., it takes only around 2 minutes to estimate parameters for a graph that contains 50 variables.
>
> **Q4:** To what extent the sufficient and necessary conditions in Sec 3 are useful for (numerical) parameter estimation in Sec 4?
>
> **A4:** Given observational data and any structure, we can always employ our method proposed in Section 4 to get an estimation of parameters. However, whether the estimated parameters are meaningful depends on the specific given structure and we need to rely on the theoretical results developed in Section 3 to answer this question. Specifically, if the given structure satisfies the sufficient conditions proposed in Theorem 1, then we can conclude that the estimated parameters are meaningful in the sense that they will be the same as the true underlying parameters asymptotically (as in this case only the true parameters can maximize the likelihood). On the other hand, if a given structure does not satisfy the necessary conditions in Corollary 1 or 2, then it can be guaranteed that the true parameters cannot be recovered (by any estimation method as there is just not enough information). In such cases, multiple (infinite number of) parameter sets can yield the same maximum likelihood. We thank the reviewer for the insightful question and have added this discussion to our revision.
>
> **Q5:** The motivation for considering edges / parmeters from observed to latent variables and between latent variables
>
> **A5:** Thank you for the insightful question. From one perspective, if we do not explicitly consider the edge coefficients from observed to latent variables and edge coefficients between latent variables, then in many cases we cannot correctly recover the edge coefficients between observed variables either. From another perspective, the recovered edge coefficients that involve latent variables are themselves practically meaningful. Take our psychometric result in Figure 4 as an example. We may be interested in how Agreeableness influences Extraversion in human personality, even though none of them can be directly observed. In this case, the edge coefficient from L3 to L2 is informative; the value of +0.39 not only indicates a positive influence but also shows that the magnitude is considerable.
>
> We genuinely appreciate the reviewer's effort and hope that your concerns/questions are addressed.
>
> [58] Toledo, Sivan. "Locality of reference in LU decomposition with partial pivoting." SIAM Journal on Matrix Analysis and Applications, 1997.

---

> > ### Author Response · Authors · 2024-08-13
> >
> > Dear Reviewer PaWo,
> >
> > Thank you once again for your insightful comments and valuable feedback. We would like to know if our response has addressed your concerns and questions? If there is any remaining confusion, we are happy to address them as soon as possible.
> >
> > Many thanks!
> >
> > Your sincerely,
> >
> > Authors of submission 11557

---

> > > ### Comment · Reviewer_PaWo · 2024-08-13
> > > **Comments**
> > >
> > > Dear Authors,
> > >
> > > thank you for your thorough answers to my questions and comments. It seems the rebuttal addresses all my concerns.

---

> > > > ### Author Response · Authors · 2024-08-13
> > > >
> > > > Dear Reviewer PaWo,
> > > >
> > > > Thank you for the positive feedback. We are so happy that all your questions/concerns were properly addressed. Your insightful review comments have helped us further improve the quality and clarity of our paper. We wonder whether you would kindly reconsider your rating based on our responses to your questions. Your consideration is highly appreciated.
> > > >
> > > > Many thanks!
> > > >
> > > > Your sincerely,
> > > >
> > > > Authors of submission 11557

---

### Official Review · Reviewer_Ljpg · 2024-07-12

**Soundness:** 3
**Presentation:** 4
**Contribution:** 3
**Rating:** 6
**Confidence:** 4

**Summary:**

This paper investigates the parameter identifiability of partially observed linear causal models, focusing on whether edge coefficients can be recovered given the causal structure and partially observed data. It extends previous research by considering relationships between all variables, both observed and latent, and the coefficients of all edges. The authors identify three types of parameter indeterminacy in these models and provide graphical conditions for the identifiability of all parameters, with some conditions being necessary. A novel likelihood-based parameter estimation method is proposed to address the variance indeterminacy of latent variables, validated through empirical studies on synthetic and real-world datasets, showing the effectiveness of the proposed method in finite samples.

**Strengths:**

The authors consider the problem of parameter identifiability in partially observed causal models with linear structural equations and additive Gaussian noise. This has not been studied in a similar way before in the literature, especially when observed and latent variables are allowed to be flexibly related. Specifically, observed variables are allowed to be parents of unobserved variables. The authors also provide graphical conditions that are sufficient for all parameters to be identifiable and show that some of these conditions are provably necessary. To address the scenario where noise covariance matrix is unknown, they propose a novel likelihood-based parameter estimation method and validate it with empirical studies on synthetic and real-world data.

**Weaknesses:**

The paper considers a restricted setting, allowing only linear relations between observed and unobserved variables and restricting the noise to be Gaussian. The theoretical guarantees hold under the scenario when the noise covariance matrix is unknown. Nonetheless, the authors mitigate this by providing an estimation scheme for the noise covariance matrix from limited data samples. The conditions of identifiability proposed are not fully necessary and sufficient; this is left for future work.

**Questions:**

I have  few questions and comments for the Authors:

*  Line 139: The authors mention that the indeterminacy of group sign is rather minor. if the parameters are identifiable only up to group sign indeterminacy, we still say that the parameters are identifiable. It would be useful to explain why group sign indeterminacy is a minor issue. It might actually depend on the underlying task, and there might be application scenarios where it is not insignificant. Some details on this would be useful in the main paper.

*  The additive noise is assumed to be Gaussian. Suppose the noise follows another continuous distribution such as Gamma or Student-t distribution. Can we use the identifiability results? If not, what is special about the Gaussian noise here compared to other continuous noise distributions?

* Referring to Proposition 1, some matrix inverses need to be calculated to compute the noise covariance matrices. Will the inverses always exist? If yes, can you elaborate on how? If no, how does that impact the application of Proposition 1?

* In the experiments section, the authors demonstrate the effect of sample size on the mean squared error (MSE) for the parameter matrix \( F \). It would be useful to also show how accurate the estimates of the noise covariance matrix depending on the sample size used.

**Limitations:**

The authors clearly describe the problem setting and assumptions. I have mentioned the main limitation of the proposed method in the weaknesses section. I don't think there are any potential negative societal impacts of this work.

---

> ### Author Rebuttal · Authors · 2024-08-07
>
> Thank you for the time dedicated to reviewing our paper, the insightful comments, and valuable feedback. Please see our point-by-point responses below.
>
> **Q1:**  Regarding group sign indeterminacy.
>
> **A1:**
> The group sign indeterminacy is rather minor with reasons as follows. (i) In practice, we can always anchor the sign of some edges according to our preference or prior knowledge in order to eliminate the group sign indeterminacy.
> For example, in Figure 4,
> if we expect that L2 should be understood as Extraversion instead of non-Exterversion, we can add one additional constraint during our parameter estimation such that the edge coefficient from L2 to E1 ("I am the life of party.") will be positive (as we believe E1 should be positively related to Extraversion).
> (ii) On the other hand, there are some application scenarios that are not influenced by the group sign indeterminacy,
> such as causal effect estimations between certain variables.
>
>
> **Q2:** What if the additive noise follows another continuous distribution?
>
> **A2:** Thank you for the insightful question. If we do not assume Gaussianity, the proposed asymptotic identifiability result still holds. The reason lies in that we only make use of the second-order statistics of the distribution and thus the additive noise can follow any other continuous distribution.
> We thank the reviewer for the insightful question and have revised to make it clear in our revision.
>
> **Q3:** Will the inverses in Proposition 1  always exist?
>
> **A3:**  Yes, they always exist.
> Thanks for asking this question which helps improve the clarity of Proposition 1. In light of your suggestion, we have updated Proposition 1 to state that all matrix inverses exist. We have added the proof to our revision with a sketch as follows.
> > *Sketch of proof.* Note that matrices $I-D$ and $I-F$ are invertible because structure $\mathcal{G}$ is acyclic. This implies $\det(I-F)\neq 0$ and $\det(I-D)\neq 0$. Define
> $$
> U=\\begin{pmatrix}
> I & 0 \\\\
> -(I-D)^{-1}C & I
> \\end{pmatrix},
> $$
> which implies
> $$
> (I-F)U=\\begin{pmatrix}
> M & B \\\\
> 0 & I-D
> \\end{pmatrix}
> $$
> and thus
> $$
> \det((I-F)U)=\det(M)\det(I-D).
> $$
> Since $\det(U)=1$ and $\det(I-F)\neq 0$, we have
> $$
> \det((I-F)U)=\det(I-F)\det(U)\neq 0.
> $$
> By the statement above and  $\det(I-D)\neq 0$, we have
> $$
> \det(M)=\frac{\det((I-F)U)}{\det(I-D)}\neq 0,
> $$
> which implies that $M$ is invertible. Similar reasoning can be used to show that $N$ is invertible.
>
> **Q4:** It would be useful to also show how accurate the estimates of the noise covariance matrix are.
>
> **A4:** We note that once the edge coefficient matrix $F$ is determined, the noise covariance matrix $\Omega$ can also be uniquely determined - we have  $(I-F^T)\Sigma_{\mathbf{V}}(I-F)=\Omega$ where the left hand side only depends on $F$ when variables have unit variance. Thus, a small MSE of $F$ usually implies a small MSE of $\Omega$. In light of your suggestion, we conducted additional experiments to empirically validate this point: under the GS case, the MSE of $\Omega$ using our Estimator-TR are 0.003, 0.001, 0.0004 with 2k, 5k, and 10k sample size, respectively. We thank the reviewer for the valuable question and have included this additional result in our revision.
>
> We genuinely appreciate the reviewer's effort and hope that your concerns/questions are addressed.

---

> > ### Comment · Reviewer_Ljpg · 2024-08-08
> > **Re.**
> >
> > Thanks for responding to questions in detail. The clarity of the paper would be improved given that the authors update the paper as they mentioned in their response to my review. I would keep my acceptance decision and score for the paper.

---

> > > ### Author Response · Authors · 2024-08-12
> > > **Author response**
> > >
> > > Thank you once again for your valuable comments which have helped improve the clarity of our paper. If you have any further questions or insights, we would be more than happy to hear from you.
> > >
> > > Thank you!
> > >
> > > Yours sincerely,
> > >
> > > Authors of submission 11557

---

### Official Review · Reviewer_WKpu · 2024-07-18

**Soundness:** 3
**Presentation:** 3
**Contribution:** 3
**Rating:** 7
**Confidence:** 3

**Summary:**

This paper introduces conditions under which DAGs can be recovered in the linear case where some nodes are observed and some are not. This DAG recovery involves computing edge weights between nodes in a causal graph. Nodes are allowed to be latent or observed, with varying types of edge weight indeterminacy depending on the structure of the causal system in question.

**Strengths:**

There is certaintly interest in estimating DAG structure in practice. The work here presents useful results for how and when that estimation may or may not occur given the structure of the causal system. The extension to latent variables is a contribution, although in practice, latent nodes might increase the already difficult task of interpreting inferred DAG structure. The misspecification analysis is useful and the presence of examples in the text helpful (with a few caveats below).

I like the title. The framing is at its strongest when the paper articulates the general conditions under which identifiability can and cannot be achieved. As for whether a given causal system in practice meets assumptions for strongest identifiability is in the end, to my eyes, a very difficult question.

Overall, the paper is a solid contribution, although I believe it could be improved (see below).

**Weaknesses:**

The text overall is generally well-written, with some caveats listed below. In my reading, the first half of the paper read more clearly than the second half. For example, I couldn't quite piece together from the discussion of estimation whether the estimated graph will be dense (all nodes connected to all other nodes, given the [seemingly?] continuous optimization being done, e.g., Eq 3. If all edges are connected to all other edges, then the relative usefulness seems to be weakened in that usually, investigators seek out a parsimonious representation of a causal system.

I was also wondering what more established methods would yield as an empirical baseline (e.g., PC algorithm); currently, the Estimator-LM (estimator with Lagrange multiplers) is articulated as a baseline. This is one of the methods introduced in the paper. An external state-of-art baseline would be most informative. The authors state that "no existing method...can achieve the same goal as ours." If this is because of the latent variable aspect, one could in principle restrict the MSE calculation to edges among observed nodes. In other words, perhaps there isn't a perfect analogue method but an imperfect comparison could be better than none. We could also get a visual comparison of the DAG among observed variables used in Figure 4 from some existing baseline methods for the Appendix.

Finally, there is little discussion of uncertainty estimation. Uncertainty estimation in the observed DAG recovery case is hard, even more so here (presumably).

There are points where the text could, to me, use more clarity in the discussion:

- Condition 1 seems relatively minimal and even intuitive. It seems very hard to know in practice if Condition 2 (line 193) holds or is even a minimal or very restrictive condition.

- I appreciate the author(s)' inclusion of Example 2 and Example 3. I think the logic could be made clear, perhaps with additional shadings or labelings that would help us see which sets of nodes and edges are doing what work regarding Condition 1 and Condition 2.

- I would make the "pretty close" language in 218 and 255 a bit more precise. Also, starting line 264. I would revise this from, "it has considerable extents of necessity, and could be expected to serve as a stepping stone towards tighter and ultimately the necessary and sufficient condition for the field." to something that also is seomwhat more precise. Also, there is no guarantee that necessity+sufficiency will be found (or perhaps there is an impossibility), so would hedge this possibility somewhat.

- Can you define what it means for "QF and F" to "share the same support" in this case? "Support" is often defined in causal inference settings as an event probability falling between 0 and 0; here, F is defined as the matrix embodying the causal edge coefficients, so I believe what is meant here is that QF and F do not share the same set of non-zero entries. Clarifying this would be helpful. I would also consider beginning with the example of indeterminacy before defining it to help the reader see your point intuitively before the formalization.

- I would definition 7 into the main text. It is an important definition used multiple times and without it, it is hard to follow the atomic cover discussion. it's also a very short definition.

Moreover, I would in general help the reader along by first explaining the concept before formally/technically defining it. Examples:

- The term structure identifiability and parameter identifiability should be clearly defined before the terms are used. (I don't think I see a clear definition before use currently; I would move the paragraph beginning on line 171 up in the text, as it is a clear articulate of the point.) In a similar vein, I would explain what atomic covers are going to do before we jump into the definition on line 164.

Other details would help this reader:

- The MSE up to orthogonal transformation is an interesting metric. Some mention of how this optimization is done would be helpful.

- I would add a sentence explaining whether GPU acceleration would or would not be helpful and why.

I also noted several minor points listed here:

- References used are inconsistent at times. Sometimes, we see reference to "condition (i)", others to Condition 1.

- Line 212. Missing space "identified upto the" should read "identified up to the". This same occurs in line 140 ("upto" should read "up to") and in other parts of the text as well.

- Line 147. "indetermincay" should read "indeterminacy". This typo occurs a few times in the text.

- Line 129. Clarify what is meant by "entails the same observation as that of..." This also appears in line 142 ("Entails the same observation"). I assume this means something about implying the same probability distribution, but helping the uninitiated reader is usually appreciated.

- Line 137. "However, if we set f1,2 = 0, then the parameters are not identifiable. These rare cases of parameters are of zero Lebesgue measure so we rule out these cases for the definition of identifiability" -> I would just say "these presumably rare cases of parameters". Probably some justification is needed to articulate why this should be rare in real causal systems. Does any prior literature speak to this?

- Line 72. I believe "the causal edge coefficient of the model" should read "the causal edge coefficients of the model".

- Capitalize "Gaussian" on line 286.

**Questions:**

- What are the implications of the diagonal covariance matrix $\epsilon_{\mathbf{V}_{\mathcal{G}}}$?

- Regarding, "As variables are jointly Gaussian, asymptotically our observation can be summarized as population covariance over observed variable". Wouldn't this statement also apply in finite samples under Gaussianity?

- A major benefit seems to be identification of edge weights. If actual interpretation of the edge weights is going to be done in practice, group sign indeterminancy would limit applicability. Would it help to anchor the sign of one edge based on prior science? Guidance?

- In Figure 4, are circulate nodes latent and nodes denoted by [LetterNumber] observed? If so, clearly articulate this in the figure label.

**Limitations:**

I see no major negative societal impacts.

---

> ### Author Rebuttal · Authors · 2024-08-07
>
> Thank you for the time dedicated to reviewing our paper, the insightful comments, and valuable feedback. Please see our point-by-point responses below.
>
> **Q1:** Whether the graph will be dense regarding Eq.3?
>
> **A1:**  We note that Eq.3 concerns the estimation of causal coefficients F, given the data and the structure G, and the entries of F that do not correspond to an edge in G are constrained to be zero during the optimization (as in lines 289-290). As the graph is given, sparsity constraints are not needed in Eq.3, which is in contrast to continuous-optimization-based structure learning methods such as NOTEARS.
>
> On the other hand, you are totally right that we do expect that the given structure is parsimonious/sparse. For example,
> if a graph contains latent variables and all variables are fully connected, then according to our identifiability theory the parameters are provably not identifiable. Roughly, a sparse/parsimonious graph has a better chance of satisfying the condition for parameter identifiability.
>
> **Q2:** Is it possible to restrict the MSE calculation to edges among observed nodes to compare with any baseline that does not allow the presence of latent variables?
>
> **A2:** Thank you for your insightful suggestion. Yes, it is possible to
> restrict MSE to edges among observed to compare with methods that do not allow latent variables.
> For example, GES with BIC scores can be considered as it estimates all the linear edge coefficients during the calculation of likelihood.
> In light of your suggestion, we conduct additional experiments to compare with GES using your proposed restricted version of MSE. Specifically, GES achieves restricted MSE of 0.41, 0.35, and 0.35 with 2k, 5k, and 10k sample size, while our Estimator-TR achieves 0.008, 0.002, and 0.001, respectively.
> It is observed that, even when we restrict the MSE to edges among observed variables, methods that do not allow the presence of latent variables cannot perform well. The reason is that,
> if two observed variables have latent confounders, we cannot correctly recover the edge coefficients among them without explicitly modeling/considering the existence of latent variables.
> We thank the reviewer again for the valuable comment and have added the additional result together with this discussion to the appendix.
>
> **Q3:** Discussion of uncertainty estimation?
>
> **A3:** Many existing works on the parameter identification problem focus on the asymptotic identifiability [18,19], and thus our theoretical analysis follows this spirit,  because this type of asymptotic identifiability result is needed before one can provide results on uncertainty estimation.
> At the same time, uncertainty estimation is certainly possible under our framework.
> To be specific, as we use maximum likelihood estimation for the parameters, some standard techniques can be readily used. For example, bootstrap can be employed to provide standard errors on linear coefficients. Chi-square test can also be done to examine the fitness of the model. Thank you for the valuable suggestion and we have added this discussion to our revision.
>
> **Q4:** What does it mean by QF and F share the same support.
>
> **A4:** Your understanding is correct. It means QF and F share the same set of non-zero entries. We have added this sentence to our revision as you suggested.
>
> **Q5:** Some mention of how MSE up to orthogonal transformation is done would be helpful.
>
> **A5:** MSE up to orthogonal transformation is calculated by solving the optimization problem in line 350. We use PyTorch with SGD to solve this problem where an orthogonal matrix Q can be directly parameterized and optimized. We thank the reviewer and have added it to our revision.
>
> **Q6:** GPU acceleration?
>
> **A6:** Yes, our optimization problem in Eq.3 is solved by gradient descent using pytorch (or any other automatic differentiation framework) and it can certainly be further accelerated by using GPU. A very related discussion can also be found in [33].
> Our current implementation is based on CPU as it is already fairly fast - it only takes around 2 minutes to handle a quite big graph that has 50 variables (as discussed in section 5.4). Thank you for the valuable suggestion and we have added this discussion to our revision.
>
> **Q7:**
> Regarding the rare cases of parameters, probably some justification is needed to articulate why this should be rare in real causal systems. Does any prior literature speak to this?
>
> **A7:** This is the typical setting in parameter identification literature where generic identifiability is concerned [5,19] (a similar spirit is shared by causal discovery literature that assumes faithfulness [48]).
> These cases are rare in the sense that they have Lebesgue measure zero.
> In real causal systems, if all the edge coefficients are randomly generated (from an absolutely continuous distribution), these cases are not a concern because they are of Lebesgue measure zero. At the same time, if edge coefficients in a causal system are deliberately or adversarially designed, then identifying these parameters using only observational data can become extremely difficult (similar to violation of faithfulness assumption in causal discovery in the sense that typical constraint-based algorithms would fail).
>
>
> **Q8:** Implications of the diagonal covariance matrix $\Omega$ of $\epsilon_{\mathbf{V}_\mathcal{G}}$
> ?
>
> **A8:** Do you mean why it is diagonal? As our framework explicitly models all the latent variables, all the
> $\epsilon_{\mathbf{V}_i}$
> are mutually independent, and thus $\Omega$ is diagonal (in contrast, the ADMG framework does not explicitly model latent variables, and thus their $\Omega$ is not necessarily diagonal).

---

> ### Author Response · Authors · 2024-08-07
> **Rebuttal by Authors Part 2**
>
> **Q9:**
> Regarding, "As variables are jointly Gaussian, asymptotically our observation can be summarized as population covariance over observed variable". Wouldn't this statement also apply in finite samples under Gaussianity?
>
> **A9:** In finite samples under Gaussianity, our observation can be summarized as the empirical covariance, which is an estimation of the population covariance.
>
> **Q10:**
> If actual interpretation of the edge weights is going to be done in practice, group sign indeterminancy would limit applicability. Would it help to anchor the sign of one edge based on prior science? Guidance?
>
> **A10:** Yes, we can always anchor the sign of some edges according to our preference or prior knowledge in order to eliminate such indeterminacy. For example, in Figure 4,
> if we expect that L2 should be understood as Extraversion instead of non-Exterversion, we can add one additional constraint during our parameter estimation such that the edge coefficient from L2 to E1 ("I am the life of party.") will be positive (as we believe E1 should be positively related to Extraversion). Thank you for your insightful question and we have added a related discussion to our revision.
>
> **Q11:**
> In Figure 4, are circulate nodes latent and nodes denoted by [LetterNumber] observed? If so, clearly articulate this in the figure label
>
> **A11:**
> Yes. We have revised the caption of Figure 4 as you suggested.
>
> Regarding the suggestions on writing such as additional shadings for examples,
> more precise statements for Remark 1,
> moving definition 7 into the main text and some explanations to the front of corresponding definitions, and correction of some typos, we thank the reviewer and have revised them accordingly.
> We genuinely appreciate the reviewer's effort and hope that your concerns/questions are addressed.

---

> > ### Comment · Reviewer_WKpu · 2024-08-12
> > **Response**
> >
> > Many thanks to the authors for their detailed responses. These answers clarify some of my questions/hesitations; the associated paper revisions should help bolster the contribution too. The discussion of sparsity and optimization will help readers understand the contribution better, as well as the impact of some of the required assumptions.
> >
> > Pondering the question of whether to alter the numerical rating, with the addition of some of the new baselines, I update my view of the paper from, "no major concerns w.r.t. evaluation" to "with good evaluation", and hence move my score to a "7".

---

> > > ### Author Response · Authors · 2024-08-12
> > >
> > > Thank you again for your helpful comments and positive feedback. It means a lot to us. We are so happy that most of your questions/hesitations were properly addressed. If you have any further insights, we would be more than happy to hear from you.
> > >
> > > Thank you!
> > >
> > > Your sincerely,
> > >
> > > Authors of submission 11557

---

### Decision · Program_Chairs · 2024-09-25

**Decision:**

Accept (poster)

**Comment:**

This paper investigates parameter identification in linear causal models including both observed and latent variables, and proposes necessary and sufficient conditions for parameter identifiability. This is an important problem that is clearly relevant for NeurIPS, and all reviewers have recommended acceptance.